# Scattered tree death contributes to substantial forest loss in California

Yan Cheng [1] ✉, Stefan Oehmcke[2], Martin Brandt [1], Lisa Rosenthal[3], Adrian Das [3], Anton Vrieling [4], Sassan Saatchi[5,6], Fabien Wagner [5,6], Maurice Mugabowindekwe [1], Wim Verbruggen [1], Claus Beier[1] & Stéphanie Horion [1] ✉

In recent years, large-scale tree mortality events linked to global change have occurred around the world. Current forest monitoring methods are crucial for identifying mortality hotspots, but systematic assessments of isolated or scattered dead trees over large areas are needed to reduce uncertainty on the actual extent of tree mortality. Here, we mapped individual dead trees in California using sub-meter resolution aerial photographs from 2020 and deep learning-based dead tree detection. We identified 91.4 million dead trees over 27.8 million hectares of vegetated areas (16.7-24.7% underestimation bias when compared to field data). Among these, a total of 19.5 million dead trees appeared isolated, and 60% of all dead trees occurred in small groups (≤3 dead trees within a 30 × 30 m grid), which is largely undetected by other state-level monitoring methods. The widespread mortality of individual trees impacts the carbon budget and sequestration capacity of California forests and can be considered a threat to forest health and a fuel source for future wildfires.

Forests worldwide are experiencing warmer temperatures and an increased frequency of severe droughts, insect outbreaks, and wildfires[1–4]. These disturbances have led to a global increase in large-scale tree mortality events, even in drought- and heat-tolerant ecosystems such as Mediterranean dry forests[1,5,6]. Severely disturbed forests can take decades to centuries to recover[2] and may even, due to regeneration failure or climatic disequilibrium, transition to an entirely different ecosystem type[7–9]. This negatively impacts forest carbon storage, biodiversity, and livelihoods of people who rely on forest resources.

California provides an extreme example of climate-induced forest declines since its 2012-2016 drought event[1]. Estimates derived from US Forest Service Aerial Detection Surveys (ADS)[10] revealed that 1–5% of the live tree biomass in 2012 died by 2017[11]. A large portion of tree deaths was a result of compound events, such as droughts, bark

beetles, and wildfires[1,12–14]. While dead trees are a natural component of ecosystems and may contribute to biodiversity[15] and growth releases of surrounding or understory trees[16], the accumulation of dead trees can lead to substantial increases in available long-burning fuels such as logs and snags, and eventually increase the occurrence of large wildfires[17,18]. The quantification of dead trees is, therefore, critical information for the optimisation of fuel reduction treatments to mitigate large wildfires and associated carbon emissions[19,20]. The current state-scale forest health surveys such as ADS[10], where surveyors sit in a fixed-wing aircraft and visually identify tree mortality, document areas with elevated mortality and the potential causes, but uncertainties remain large in the estimates of the extent and number of dead trees[21,22].

To better identify the actual extent of tree mortality, it is important to precisely locate stressed or dead trees. Currently, most

[1]Department of Geosciences and Natural Resource Management, University of Copenhagen, Copenhagen, Denmark. [2]Department of Computer Science, University of Copenhagen, Copenhagen, Denmark. [3]US Geological Survey, Western Ecological Research Center, Three Rivers, Sequoia and Kings Canyon Field Station, Three Rivers, CA, USA. [4]Faculty of Geo-Information Science and Earth Observation (ITC), University of Twente, Enschede, The Netherlands. [5]University of California, Los Angeles, CA, USA. [6]Jet Propulsion Laboratory, California Institute of Technology, Pasadena, CA, USA. ✉e-mail: yach@ign.ku.dk; smh@ign.ku.dk

tree-level mortality data are collected through ground surveys. Despite providing detailed information, the spatial coverage of these surveys is limited by accessibility, time, and cost. Near-surface remote sensing technologies, such as UAV- and airborne-based optical and LiDAR, have recently been explored as an alternative method for fine-resolution forest health monitoring from regional to landscape scales[23–25]. Nevertheless, these data are collected on a need basis and do not provide wall-to-wall coverage at the state scale. Systematic assessment of forest degradation at regional to landscape is possible with satellite images such as those from Sentinel-2 and Landsat[26–33]. Finer resolution images such as PlanetScope, WorldView, and Pleiades have also been tested to map tree mortality[34–36]. Restricted to area-level estimates via vegetation index anomalies[26–29,33–36], biochemical properties[30], or by mapping deadwood fractions at the pixel level[31,32], these approaches can not readily count individuals and likely miss scattered dead trees[37]. Consequently, the actual counts of tree mortality and the contribution of small groups of dead trees over large areas remain unknown. Assessments at the tree level based on finer resolution data are therefore needed across broad extents. The National Agriculture Imagery Programme (NAIP)[38] provides imagery at sub-metre resolution for the entire U.S. during the growing seasons on a biennial basis. Advanced computer vision algorithms create the opportunity to effectively apply such imagery for fine-resolution and large-area mapping of tree mortality[39–41].

In this study, we used deep learning to map individual dead tree crowns in California from 2020 NAIP images. We defined dead trees as overstory standing deadwood or snags. The uncertainty of predicted dead tree density was assessed with ground observations at the tree and plot levels. We studied the main patterns, such as spatial distribution, species composition, and damage agents, emerging from our wall-to-wall tree mortality mapping in California. We quantified the contribution of isolated and small groups of dead trees within 30 × 30 m grids to the total extent. From the individual dead tree map, we derived the crown size of each dead tree and crude approximations of the recency of mortality by classifying the mortality at "brown" or "grey" stages based on the crown colour. By combining the count of dead trees, the percentage of brown-stage mortality, and the median size of dead crowns per ha, we mapped tree mortality hotspots that suggest massive and recent loss of forests. Lastly, we presented an example of multi-year tree mortality mapping by applying the model trained for 2020 directly to NAIP images acquired in adjacent years over a subset of the study area and evaluated the accuracy against field observations. We identified 91.4 million dead trees from 2020 NAIP images with 19.5 million appearing solo within 30 × 30 grids. Bark beetles and fires are the primary damage agents according to the aerial and ground surveys. Multi-year mapping unravels diverse changes in dead tree density despite inconsistent accuracy attributed to geometrical and spectral variations in NAIP images from different years. Our results highlight the need and feasibility to map and characterise tree mortality at the individual tree-level over large extents. The tree mortality maps provide detailed information for forest management and the understanding of the mechanisms of climate-induced tree mortality.

## Results

### Mapping of individual and scattered dead trees in California for 2020

We trained a deep learning model on about 24,000 manually digitised dead trees to detect individual dead tree crowns in vegetated areas in California from 7645 NAIP aerial images in 2020[38] (Methods). When comparing to an independent set of about 3000 digitised dead tree crowns, we found an overall underestimation bias of 3.61% and a Mean Absolute Error (MAE) of 2.27 dead trees ha-1 (Supplementary Fig. 1). The overall underestimation bias and MAE for fire-related mortality (15.3% and 4.56 dead trees ha-1) were greater than non-fire-related

mortality (0.15 and 1.91 dead trees ha-1; Supplementary Fig. 2). When compared to point locations of dead tree mapped in field surveys between 2016 and 2020 (Methods), the underestimation bias was 16.7-24.7% (Supplementary Table 1). The comparison to the count of dead trees with a diameter at breast height (DBH) > 40 cm for plots visited in 2016 and 2018 (Methods) indicates an underestimation bias between 5.08-19.9% and an MAE of 2.2–2.9 dead trees ha-1 (Supplementary Fig. 3). Some of the omission errors can be explained by the presence of severely decayed trees with a short trunk and/or a small "crown" that can be covered by adjacent tree canopies, especially in dense canopies (Supplementary Fig. 4b) and/or in images with severe distortion issues due to off-nadir view angles (Supplementary Fig. 4a, c). In addition, given that noticeable decay of standing dead trees usually starts within 2–5 years post mortality[42], dead trees recorded before 2020 may have fallen down and therefore could not be detected from aerial images acquired in 2020.

We found about 91.4 million dead trees over 27.8 million hectares of vegetated areas (Methods) in California, which accounts for 2.33% of the estimated tree count in 2011[43]. Our report on the total number of dead trees is smaller than previous estimates (156.7 million) from ADS 2012-2020[10] (Methods; Supplementary Table 2). However, our model appears to capture a larger number of dead trees in Northern California as compared to other sources (Supplementary Fig. 5). While the time mismatch and the exclusion of dead trees fallen before 2020 in this study could partly contribute to discrepancies between sources, ADS' estimates were found to substantially over-estimate the actual dead tree count due to mapping design limitation (hand-drawn polygons)[21].

Among 16 main forest-type groups[44], California mixed conifer appeared to be the most affected, accounting for half of all detected dead trees (Fig. 1b) and the highest percentage of mortality per ha (Fig. 1c). The western oak group was the next most vulnerable forest type (Fig. 1b), confirming recent studies based on field observations[45]. Geographically speaking, the North Coast and central and southern Sierra Nevada were the most affected regions. A total of 8.22% of the dead trees were found in Shasta-Trinity National Forest (NF) followed by Klamath NF (7.51%) and Sierra NF (7.36%); these NFs account for 2.03%, 3.53%, 5.44% of the trees mapped in 2011[43], respectively. Shasta-Trinity NF has the highest percentage of tree mortality since 2011 among 17 national forests in California (Fig. 1a).

Tree mortality may be attributed to single or multiple damage agents[12–14,22] depending on the interactions between damage agents such as a combination of biotic and abiotic agents or multiple species of pests or diseases. In particular, the prolonged droughts in California since 2012 have been suggested as one of the underlying triggers of bark-beetle-related tree mortality[14]. By overlapping the individual dead tree map with damage agent surveys[10] and fire perimeters[46] from 2012 to 2020 (referred to as damage agent database hereafter, Methods), we assigned potential damage agents for 67.7% of the detected dead trees. As shown in Supplementary Fig. 6a–c, biotic agents such as bark beetles (42.5%), cankers (1.5%), wood borers (0.89%), and combinations of bark beetles and wood borers (0.89%) were the dominant damage agent of tree mortality in California, accounting for 45.8% of total dead trees detected, followed by fires (27.2%), drought (16.5%), and a combination of bark beetles and fires (7.1%). The majority of biotic agent-related tree mortality was found in conifer-dominated woodlands. Fires occurred throughout California and affected all forest types. Single-agent drought-related mortality was mostly observed on oak woodlands in the foothills of central California. Wild animals, such as wild boar, and human activities, such as herbicides, were found to have a minor impact on tree mortality (1.2% and 0.33%; Supplementary Fig. 6c).

By integrating the classic watershed algorithm in post-processing (Methods), our model was able to separate clumped dead trees, providing robust count estimates in different scenarios, such as scattered

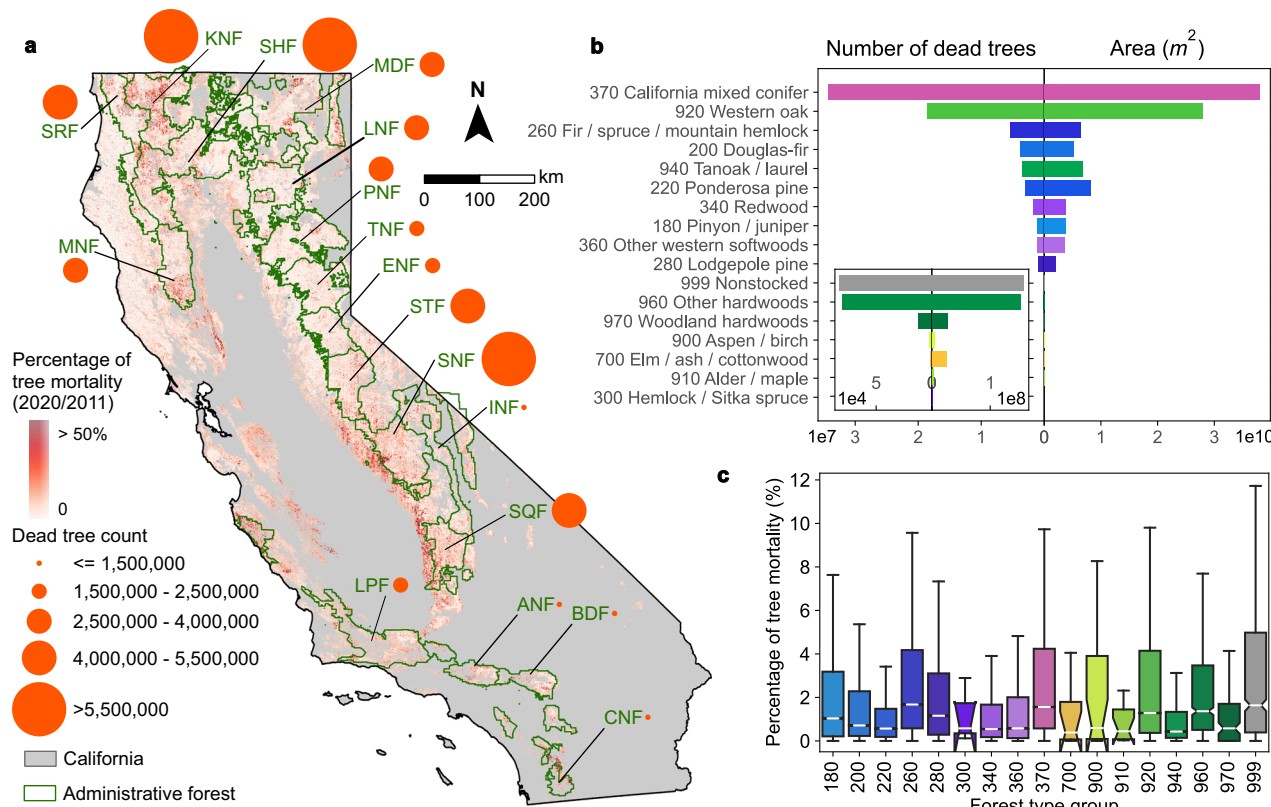

**Fig. 1 | Status of tree mortality in California from individual dead trees detected from NAIP aerial images in 2020.** In panel **a**, the orange circles denote the total count of dead trees in each national forest. Supplementary Table 3 provides the lookup table for national forest abbreviations. The underlying map represents the percentage of tree mortality, which is the count of detected dead trees against the count of all trees in 2011 within 240 × 240 m grids[43]. Only forests, shrublands, and grasslands that are contained in the National Land Cover Database 2019[63] and ESA WorldCover 2020[64] are included in the mapping (Methods). **b** Total number of dead trees and spatial coverages for 16 main logical ecological groupings of forest types in California[44]. **c** Box plots of percentages of tree mortality per ha for each forest-type group. The boxes represent the interquartile range (IQR) which is between the 25th and the 75th percentile of the percentages of tree mortality. The whiskers represent 1.5 times the IQR. The white lines inside the boxes represent the medians. The notches inside boxes represent the 95% confidence intervals for the medians. Random selection of 30% of the pixels per forest-type group was applied to mitigate the spatial auto-correlation. The colour scheme used in panels **a**, **b** is consistent with the forest-type group map (Supplementary Fig. 7), representing different forest-type groups.

and clustered die-offs in open and closed canopies (Fig. 2a–c). To quantify scattered die-offs, which are likely to be missed by conventional monitoring methods, we aggregated the individual dead tree map into a count map at 30 m resolution and found that 19.5 million dead trees appeared solo within 30 × 30 m grids (Fig. 2d), accounting for 28.4% of predicted dead trees without bias correction (Methods). Smaller groups of dead trees (≤3 dead trees within a 30 × 30 m grid) account for about 60% of the total count of dead trees (Fig. 2d). Compared to a widely used forest change map for 2000–2020 derived from Landsat images at 30 m resolution[27], 71% of the dead trees detected in our study were not captured by this dataset, among which more than half (56.6%) of the dead trees are in 30 × 30 m grids with one (34.4%) or two dead trees (22.2%; Fig. 2e). While the Landsat-based forest change map is not tailored to capture small-scale mortalities[27], it performs well in detecting large mortality clusters from fire and bark-beetle outbreaks (Supplementary Fig. 8a and Supplementary Fig. 6a) and general trends in mortality over time (Supplementary Fig. 8b). Nevertheless, the Landsat-based maps miss out on isolated and scattered dead trees, which, as we show, make up for a significant part of the total mortality in California, highlighting the importance of mapping tree mortality at the tree-level towards improved estimates of forest loss.

## Beyond dead tree localisations

From the individual dead tree map, we calculated additional structural metrics and the mortality stage and aggregated them into hectare grids (Methods). The additional structural metrics consisted of dead tree count, median dead tree crown size, and dead canopy area per ha (Supplementary Fig. 9a, b, e). The median of dead tree crown size per ha can be used to understand the decay stages and tree sizes/diameters in combination with ancillary information such as species[47]. The dead canopy area and median dead tree crown size come with an uncertainty factor (Supplementary Fig. 9f), which we quantified by assessing the geometric distortions in NAIP images caused by off-nadir view angles (Methods).

The mortality stages such as the brown-stage and grey-stage are determined by the colour of dead tree canopies and imply the existence of foliage, which can be a proxy for the recency of mortality when comparing within the same species group. Brown-stage trees appear brownish or reddish on RGB images (Fig. 3e) and imply recent tree death with dried or decoloured foliage remaining on stems[48]. Grey-stage trees have lost their foliage, appear greyish or whitish (Fig. 3f), and imply long-standing dead woods[48]. Differentiating between brown- and grey-stage allows for crude approximations of recent mortality from single-year NAIP images, along with species and damage agent information. The percentage of brown-stage dead trees against the number of all dead trees (Methods) indicates areas with recency of mortality, which is relevant for indicating a potential ongoing mortality event, such as a burgeoning bark-beetle outbreak. This indicator is also relevant to fire risk modelling and fire behaviour forecasting because brown-stage trees are more likely to increase the risk for crown fires as compared

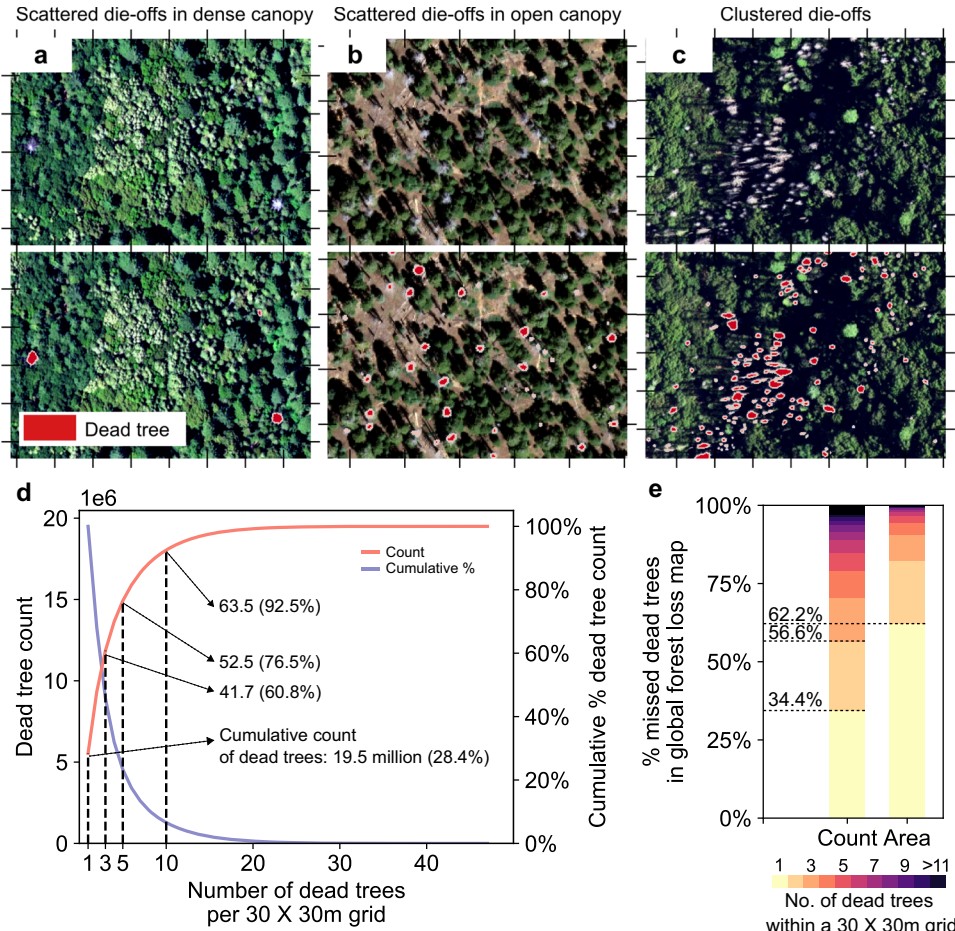

**Fig. 2 | Contribution of isolated and scattered dead trees. a–c** examples of individual dead tree segmentation in multiple scenarios. In each panel, the upper rows are the NAIP true colour images, while the bottom rows present dead tree crowns (red shapes) in predictions. The ticks around each frame represent 30 × 30 m grids. **d** total count and cumulative percentage of total dead trees for 30 × 30 m grids with a number of dead trees ranging from 1 to 47. The cumulative counts of dead trees were derived directly by aggregating the individual dead tree segmentations without applying bias correction (Methods). **e** Percentage of tree mortality (count and area) grouped by number of dead trees within a 30 × 30 m grid for areas that were not mapped in Global Forest Change v1.9 (2000–2020)[27].

to grey-stage dead trees, due to the presence of flammable low-moisture content foliage[17,49,50].

In total, 19.1% of all identified dead trees were classified as being in the brown-stage mortality. The most affected forest group types were California mixed conifer, western oak, and fir/spruce/mountain hemlock with 18.1%, 18.9%, and 32.5% of dead trees classified as brown-stage, respectively (Supplementary Fig. 10a). Among all forest group types, Lodgepole pine, fir/spruce/mountain hemlock, and pinyon/juniper had the highest percentage of brown-stage mortality, accounting for 39.5%, 32.5%, and 28.3% of the total count of dead trees in each group (Supplementary Fig. 10a). These three groups also occupied only a relatively small spatial extent, especially for Lodgepole pine, which made up for 0.97% of the forested areas in California according to ref. 44.

We selected three independent metrics to characterise tree mortality at the landscape scale: count of dead trees, percentage of brown-stage mortality, and median of dead tree crown size per ha. By combining these metrics in the RGB colour space (Fig. 3), three types of mortality hotspots stood out: first, newly developed mortality (stressed/dead trees with dried or decoloured foliage), denoted in greenish colours (Fig. 3c), refers to areas with high percentages of brown-stage mortality; second, massive dead canopy areas, denoted in magenta colours (Fig. 3b), represent areas with large numbers of dead trees and large dead tree crowns; third, large-scale legacy mortality (Fig. 3a), denoted in reddish colours, implies areas with a large amount grey-

stage mortality with small dead tree crowns. Hotspots of recent mortality (high percentage of brown-stage dead trees) were mainly concentrated in Sierra Nevada Range (green and yellow in Fig. 3), which is likely due to increased droughts and bark-beetle outbreaks[14,25]. In contrast, the Klamath Mountains/California High North Coast Range where a large count of grey-stage dead trees in relation to historical fires was uncovered (red to magenta in Fig. 3).

## Multi-year tree mortality mapping

We directly applied the model trained using NAIP images in 2020 to the NAIP images from 2016, 2018, and 2022 over a spatial subset of the study area. The model was able to detect the most recent mortality from NAIP 2022 (Fig. 4d), the gradual increase in mortality from 2016 to 2022 (Fig. 4e), and the decreased number of dead trees due to manual removal, natural falling or decomposition (Fig. 4f). Figure 4d, e illustrate some of the inconsistencies in between-year image distortions and geo-referencing of NAIP imagery, which introduces challenges for comparing changes at the individual tree level. However, this challenge is reduced when aggregating to count of dead trees per ha (Fig. 4c), which effectively shows temporal trends of tree mortality over the test area. The spatial-temporal patterns of tree mortality (Fig. 4d–g) provide detailed information for future studies in drivers and help improve the early warning of tree mortality.

By comparing to field surveys of dead trees or snags from 2016 to 2023 (Methods), we found an underestimation bias between 16.7% and

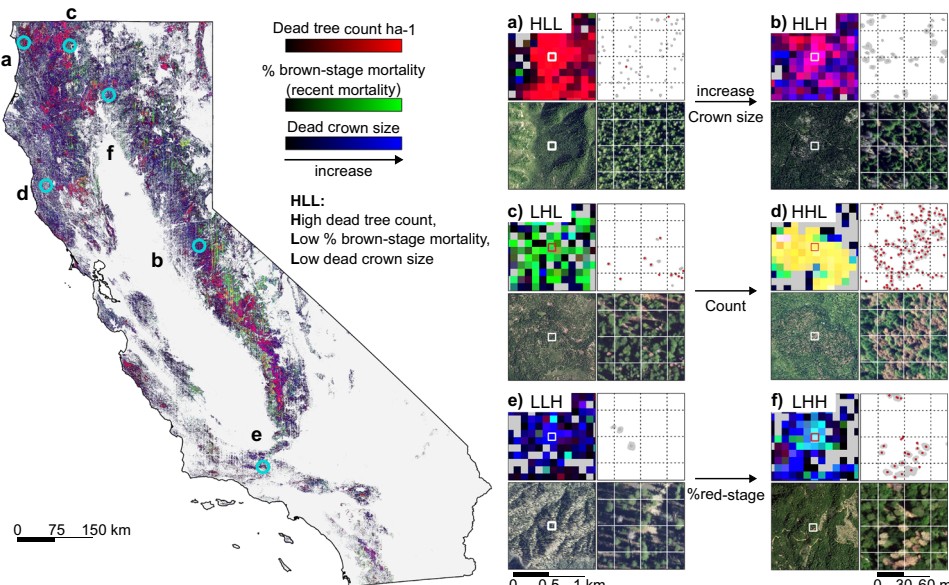

**Fig. 3 | Characterisation of tree mortality at the landscape scale based on structural metrics and mortality stages of individual dead trees.** The left-hand map shows three metrics of dead trees in RGB colour space (referred to as 3-prong mortality map hereafter), i.e., count of dead trees per ha (Red), percentage of brown-stage mortality per ha (Green), and median of dead tree crown sizes per ha (Blue). All three bands are normalised to values between 0 and 1 using the 2nd and 98th percentile of all hectare grids in California as the minimum and maximum values, allowing to highlight of relative highs and lows within California. Separate maps for each of the three metrics are provided in Supplementary Fig. 9. Panels **a**–**f** are examples of six generic types of tree mortality, as characterised based on dead tree count (first character: H=high, L=low), % brown-stage mortality (second character) and dead tree crown size (third character): **a** HLL, **b** HLH, **c** LHL, **d** HHL, **e** LLH, **f** LHH. Each panel consists of four items. The top left item is a sample area from the 3-prong mortality map. The top right item shows individual dead trees detected within a one-hectare grid located at the centre of the sample area. The crown area of the dead tree is shown in grey. The red dot indicates brown-stage mortality. The dotted lines represent 30 × 30 m grids. The lower left and lower right panels show the true colour NAIP images corresponding to the upper left and upper right panels. The geolocation of each sample area is indicated by a cyan circle on the 3-prong map.

53.1% (Supplementary Table 4). The underestimation biases for 2016, 2018, and 2022 are higher than for 2020, which could be a result of different geometrical and spectral characteristics of the 2016, 2018, and 2022 images with respect to the 2020 images that were the basis of the training samples. Further addition of training samples for the different years may improve the accuracy of multi-year mapping. The highest bias was found in predictions for 2022 when compared to field observations before 2021, which could be partially explained by the fire disturbances that occurred in 2021 over the field sites.

## Discussion

Standing dead trees are a natural component of any forested ecosystem and play an important role in supporting biodiversity[15] and mediating the growth trajectories of surrounding trees[16]. However, climate change and increasing climate extremes, have globally pushed forested ecosystems to their limit of safe functioning, sometimes, resulting in large-scale die-offs[2]. In this study, we applied semantic segmentation coupled with the Deep Watershed algorithm to locate and characterise overstory standing dead trees from sub-metre resolution aerial photos at the California state level. We showed that 60% of the dead trees are isolated or in small groups of two to three, and would have remained unseen at 30 m Landsat resolution[27], which emphasises the importance of mapping individual dead trees at large scales for the accurate count and area estimates of tree mortality.

Systematically assessing tree mortality on a regular basis is needed to monitor forest dynamics and disentangle when tree mortality appears to deviate from normal succession and functioning of forest ecosystems[2]. Indeed, unhealthy or dead trees can also facilitate insect outbreaks which may eventually lead to large-scale forest mortality[12,51]. Applying the model to map individual dead trees for multi-year NAIP images over a test area, we showed that a clear trend can be identified in the count of dead trees per ha in spite of inconsistent performances

across years. Given the availability of NAIP images, this demonstrates the potential to map tree mortality for the contiguous United States on a biennial basis while accurately accounting for scattered dead trees. We also explored the dominant damage agents of tree mortality in California from 2012 to 2020 by overlapping dead tree maps with ancillary datasets of damage agents documented through aerial and ground surveys. We found that bark beetles and fires are related to nearly 70% of the dead trees mapped in this study. Despite uncertainties in the ancillary datasets[22], the attribution of damage agents at the tree-level provides spatially detailed information supporting future studies on spatial patterns of damage agents. Together with the temporal dimension, this enables holistic profiles of existing mortality at fine spatial resolution, directly supporting the monitoring of forest health for large extents and the understanding of the interaction between climate and biotic stressors[52]. In addition, similar deep learning frameworks have been used to map live trees from high-resolution satellite and aerial images[39–41], which in combination with the tree mortality maps enables the study of tree regeneration after forest degradation.

Dead tree biomass estimates are also a critical input for modelling fire risk[35] and forest carbon fluxes[18,53]. Standing brown-stage dead trees increase fire risk given their low fuel moisture and thus stimulate crown fires. Containing some of the highest densities of carbon worldwide[54], wildfires can turn California's forests into a net emitter of carbon[20]. The localisation and mortality stage classification of individual dead trees presented in this study therefore provides spatially exhaustive information on large ground fuel loads, which overcomes the challenge in empirical model-based extrapolation attributed to the weak and varied correlations of certain fuel classes to environment variables[19]. As fire spread is strongly related to spatial heterogeneity of fuel density and fuel flammability at fine resolutions[17,55], our mapping is expected to improve fire risk and fire behaviour forecasts. Deadwood

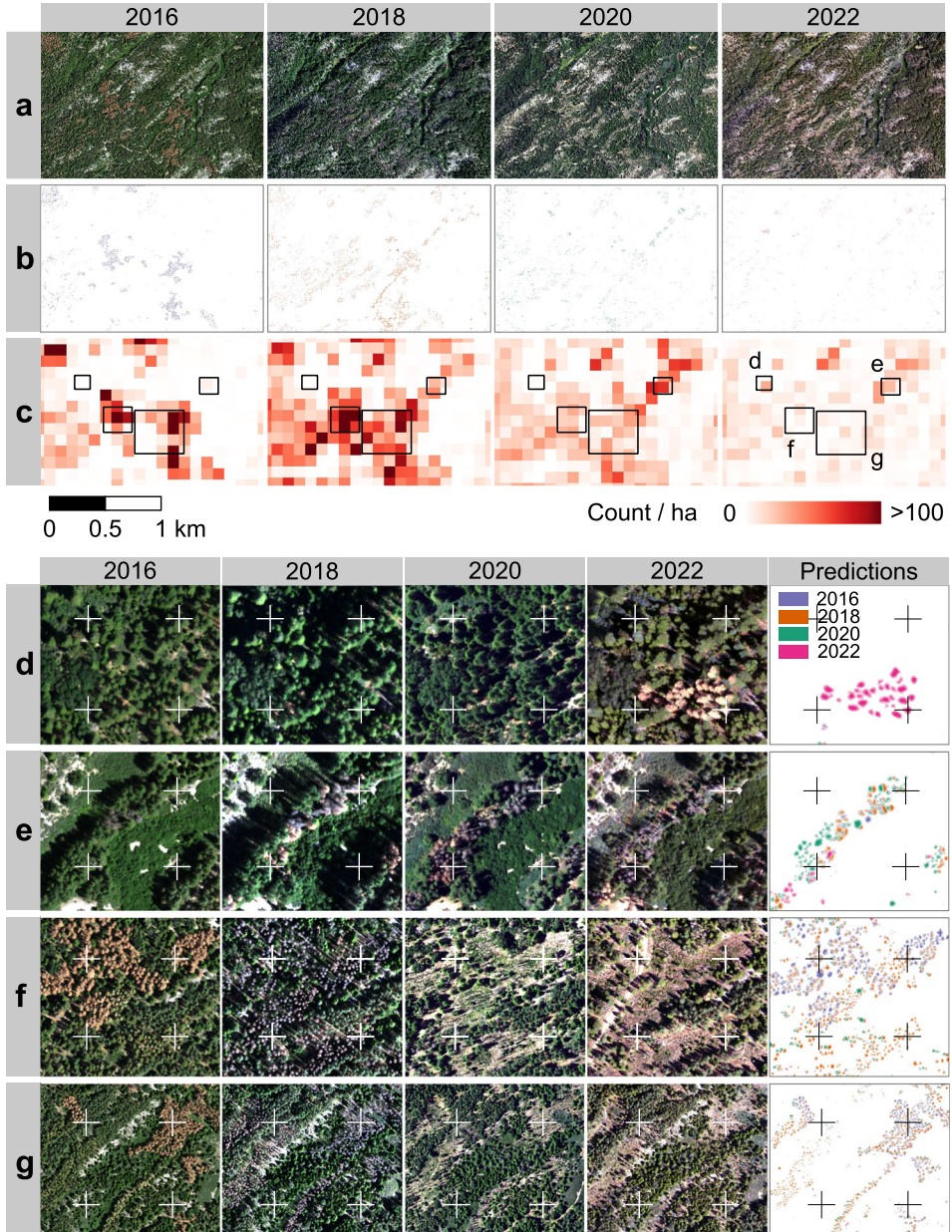

**Fig. 4 | Tree mortality mapping from multi-year NAIP images (2016, 2018, 2020, and 2022) for a spatial subset of the study area. a** NAIP image from 2016–2020. **b** predictions of individual dead trees. **c** count of detected dead trees per ha. **d**–**g** detailed views of multi-year NAIP images and detected dead trees: **d** most recent mortality from NAIP 2022; **e** increased tree mortality from 2016 to 2022; **f** decreased number of dead trees detected due to the manual removal of dead trees or the natural decomposition/falling of dead trees; g, spreading of tree mortality from the north to the south. The locations of these four detailed views are indicated with black bounding boxes in panel **c** for 2022. The crosses in panels **d**–**g** are for cross-referencing the same locations in scenes from different years.

carbon stocks are estimated to be ~8% of global forest carbon stock[53,56], but the uncertainty from current estimates is high. Research as to how mortality metrics, such as those estimated from individual dead tree polygons, can improve fire risk modelling and carbon accounting is still needed, but there is high potential for incorporating dead tree information, as obtained in our study, to improve the accuracy of such applications.

Our results also highlight several uncertainties. First, the ground observations are geographically limited to the Sierra Nevada Range and species-wise dominant by conifers; therefore, they may not represent the entire California. Despite that, the model was also evaluated against manually digitised labels across California that cover diverse species groups and elevation ranges (mean ± std.: 1,539 ± 916 m; Supplementary Fig. 7). Nevertheless, standardised data collection and data sharing

protocols for tree mortality are necessary to facilitate more comprehensive evaluation against ground observations. Secondly, despite a relatively high accuracy for large overstory dead trees, Supplementary Fig. 3d-f presents a large underestimation bias when including dead trees of all decay and DBH classes, such as for severely decayed snags and understory dead trees. As shown in the ground observations used in this study (Supplementary Table 5), there were only 41.6% (± 19.7%) of dead trees belonging to medium or tall classes (≥15 m in height estimated from DBH using species-specific allometry equations[57]). Higher resolution drone or/and aerial imagery[32] and active remote sensing technologies such as Light Detection and Ranging (LiDAR) could compensate for the mapping of small understory dead trees[58,59]. Third, the geometric distortion in NAIP aerial images can lead to the omission of dead trees with small crowns shaded by surrounding

canopies (Supplementary Fig. 4) and thereby introduce uncertainties in the estimates of dead tree crown size. Here we have limited the impact by not considering dead trees exceeding an eccentricity of 0.8 (Methods). Further correction of dead tree crown size would require the disclosure of metadata of raw NAIP images such as view angles and flight height. Fourth, tree mortality can be a result of compound events and interactions between multiple damage agents[1,12–14,22,52], which has proven to be challenging to document exhaustively through aerial or ground surveys, including the data used in this study (ADS)[22]. In particular, it is difficult to attribute drought especially when compounded with other damage agents such as bark beetles or fires. Nevertheless, we provide the first tree-level analysis of damage agents at the state scale, which can serve as first-hand material to understand the spatial and temporal patterns of tree mortality related to different damage agents.

In conclusion, our wall-to-wall mapping and characterisation of individual dead trees at California state-level provide detailed and actionable information for forest health assessment and management. The fine-resolution mapping may also support improved carbon accounting as well as fire risk and behaviour modelling. Further, repeated mapping combined with ground observations can bring insights into the mechanisms of climate-induced tree mortality and contribute to early warning systems in areas that may face widespread forest mortality in a changing climate.

## Methods

### NAIP aerial images

NAIP data consists of sub-metre to metre resolution optical images that have been acquired every one to two years during the growing season at pan-US scales since 2003[38]. The image quality and geo-referencing accuracy vary between years, depending on the hardware specifications and campaign conditions. Since 2016, the campaigns have employed a better optical sensor with red (619–651 nm), green (525–585 nm), blue (435–495 nm), and near-infrared (808–882 nm) bands. The spatial resolution has increased from 1 m to 60 cm, and the geo-referencing accuracy (RMSE) was reported as 6 metres[38]. In this study, we downloaded 7645 NAIP tiles (>3.4TB) acquired in 2020 from Google Earth Engine to have complete coverage over vegetated areas (excluding agricultural areas) in California. These images were taken from the middle of April to early August (Supplementary Fig. 11), thus allowing a clear distinction between dead trees and deciduous trees.

### Ancillary datasets

The list of ancillary datasets and their sources are in Supplementary Table 6. The Woodland for California dataset[60], California border[61], and the ecoregion map[62] were used to preselect the NAIP tiles for Central Valley and Mojave Desert (Supplementary Fig. 11), given that the tree cover for these two ecoregions is less than 5%. For the other ecoregions, we downloaded all available NAIP images from 2020. USA National Land Cover Database 2019[63], ESA WorldCover 2020[64], cities[65], urban[66], and water shapefiles[67] were used to mask non-vegetated and agricultural areas in all our analyses. The tree count map[43] depicts the total number of live trees per acre in 2011, which was defined as the initial status of live tree count before the 2012-2016 drought event and used to calculate the percentage of trees that died between 2011 and 2020. The Forest-Type Group[44], which depicts dominant species within 240 × 240 m grids, was used to unravel tree mortality across forest-type groups. This dataset was created by interpolating plot data collected between 2014 and 2018. The California forests consist of 16 forest-type groups, with California mixed conifer as the dominant group (Supplementary Fig. 7). To identify the most affected national forests, we used Administrative Forest Boundaries[68] and only considered national forests with more than 90% of the area inside the border of California which includes in total 17 national forests (Supplementary Table 3).

We used Global Forest Change v1.9 (2000–2020)[27] as the representative of satellite imagery-based forest disturbance maps to illustrate potential misses of dead trees mapped at pixel levels. This dataset was generated from the time series of Landsat imagery at 30 m resolution with information on the year of forest loss. It is important to note that ref. 27 defines trees as all vegetation taller than 5 m in height, and forest loss as a stand-replacement disturbance. The product is therefore not tailored for monitoring forest degradation due to selective logging or small-scale mortality.

To study the potential damage agents of tree mortality, we used surveys from ADS[10] which have been conducted on a yearly basis since 2003 in California. These surveys consist of rough extents or point locations of recent mortality (brown-stage mortality) identified by observers from a fixed-wing aeroplane at approximately 1500 feet above the ground. Each identified mortality area contains information on potential damage agents based on expert knowledge and ground reports. We used ADS data collected between 2012 and 2020, which covers about 6.1 to 19.6 million hectares of forested land in California. The 2020 survey was done by visual inspection from very high-resolution images (0.25 to 0.6 m) and only covered ~10% of the usual survey area due to the suspension of the flight campaign. Therefore the total number of dead trees from ADS2020 does not represent the entire California. As ADS polygons mainly represent areas with non-fire-related tree mortality, to assess fire-related tree deaths, we employed fire perimeters produced by CAL FIRE[46] and fuel disturbance data from LANDFIRE[69].

In addition, the individual dead tree count was compared to the snag density map for 2021[70] for northeastern California and southern Sierra Nevada. The snag density map was extrapolated from point plot observations using geospatial datasets[70], and depicts the number of standing dead trees with a DBH larger than 20 inches (~50.8 cm) for all species and all decay classes within 30 × 30 m grids[70]. To align the timeframe between our product and the snag density map, areas overlapping with fire perimeters for 2021[46] were masked out. Nevertheless, non-fire-related tree mortality in 2021 could still explain part of the discrepancies between products.

### Deep-learning-based Individual dead tree detection

We employed instance segmentation to detect and characterise individual dead trees for all NAIP images used in this study. To that end, we used NAIP images as input to an adapted convolutional neural network. We evaluated the location of dead trees and their count per ha against visually interpreted dead trees from NAIP images and ground observations. For the evaluation of dead tree crown size, we compared the model output with manually digitised dead trees from NAIP images. The model architecture and model assessment are described in the following sections and Supplementary Fig. 12.

**Label data preparation.** We sampled 714 patches with a size of 256 × 256 pixels from the NAIP images (~1685 hectares) overlapping with the sample areas selected for the ADS visual inspection of tree mortality in 2020[19] to include diverse geographical conditions and damage agent categories (Supplementary Fig. 7 and Supplementary Fig. 6). A total of 87 patches were used as control (background class in the model). Those patches were either located over non-vegetated areas or characterised by the absence of tree mortality. We digitised all dead trees within all patches, which consisted of ~27,000 dead trees. The patches were then split into a training, validation, and test set in the ratio of 8:1:1.

**EfficientUNet architecture.** We customised the well-known UNet architecture for image segmentation to map individual dead trees in California from very high-resolution aerial imagery. UNet was introduced for cell segmentation in biomedical images[71] and has proven its capability in tree detection from optical imagery[39–41]. The customised

architecture uses EfficientNet[72] as the encoder and applies batch normalisation after each layer. Going forward, we will call it EfficientUNet. The network was trained with NAIP images as input and visually interpreted dead trees as labels to learn from. To simulate image conditions and avoid model overfitting, image augmentation, including geometric and spectral transformations, was performed for 20-50% of the total number of input patches. Specifically, we applied an affine transformation to simulate the noticeable geometric distortions in NAIP images over mountainous areas. The Focal Tversky loss function[73] together with the Adam optimiser was used for model parameter optimisation. We set alpha, beta, and gamma values in the Focal Tversky loss function as 0.4, 0.6, and 2. When the gamma value is greater than one, the focal loss part in the loss function is activated, which has proven a good performance with imbalanced datasets[73]. When the alpha value is larger than the beta value, the model focuses on minimising false positive predictions[73], which results in relatively conservative predictions for dead trees.

**Deep Watershed algorithm.** One of the challenges in image segmentation is to separate touching objects. For example, in dense canopy areas, tree crowns can overlap with each other, which results in clumps in the prediction and leads to an underestimation of the count. To separate clumped canopies of dead trees, we applied an adapted version of the deep watershed algorithm[74]. Applying the deep watershed algorithm results in energy maps that represent the increasing distance of a pixel to its nearest boundary with higher energy levels (Supplementary Fig. 12b). These energy level maps can then be used as an input to the Watershed algorithm[75], which will return separated instances (e.g., individual dead trees). In our adaptation, we replaced the usual classification of energy levels (e.g., with softmax activation), which assumes independence between classes, with an ordinal classification[51], where we need to predict the presence of low energy before higher energy levels can be predicted. This adaptation assures that (1) the energy levels represent the distance from the border, which results in an intuitive and meaningful label of the predicted segments, (2) no unrealistic holes remain within the predicted dead tree segments, (3) tree object boundaries can potentially be sharpened or refined during postprocessing by removing lower energy levels, which would be a more informed action than simple shrinking, (4) consistency is required for easy separation of closely grouped instance (e.g., a change in the direction of energy level gradient serves as an indicator of a new instance).

**Model training and prediction.** We trained a model with about 24,000 manually digitised dead tree polygons selected across the study area. The target outputs consist of an energy map and an instance band (Supplementary Fig. 12a). The instance band was created from the energy map using the classic watershed algorithm with a connectivity of 2, where each instance is denoted as a unique integer value. The instance here refers to the predicted dead tree crown. In a post-processing step, we counted the instances within hectare grids. At the end of each training iteration, we calculated the MAE of dead tree count per patch for the validation set as:

$$MAE = \frac{1}{n} \sum_{i=1}^{n} \left| Y_i^{obs} - Y_i^{pred} \right| \qquad (1)$$

where $Y^{obs}$ is the labelled or observed count per patch or plot, $Y^{pred}$ is the predicted count per patch or plot, $n$ is the total number of patches or plots. Unless stated otherwise, the definition of these variables remains the same for other equations hereafter.

We determined the best-performing model as the one with the lowest MAE (Eq. (1)) of dead trees per patch. With that, the model was tuned to minimise the MAE of count per patch, meaning optimising predictions at the patch level[76]. However, when considering all the patches as a whole, the low MAE may not imply a low relative total error[76] (rTE) calculated as:

$$rTE = \frac{\left| \sum_{i=1}^{n} (Y_i^{obs} - Y_i^{pred}) \right|}{\left| \sum_{i=1}^{n} Y_i^{obs} \right|} \qquad (2)$$

To mitigate the propagation of systematic errors at the patch level to the sum over large areas (many patches), we followed ref. 76 and applied bias correction for the count maps by adding the Mean Error (ME) of count per grid area calculated for the training and evaluation set as:

$$ME = F \times \frac{1}{n} \sum_{i=1}^{n} (Y_i^{obs} - Y_i^{pred}) \qquad (3)$$

where $F$ is a scale factor that converts the patch or plot size to one hectare.

Unless stated otherwise, all dead tree counts aggregated from the individual dead tree map are bias-corrected. For tree-level or subset-level analysis, for example, when comparing predicted dead tree density at 30 m resolution to Landsat-derived forest loss data, the count of dead trees for each 30 × 30 m grid was directly derived from individual dead tree segmentations without bias correction.

**Model performance.** The model performance was evaluated in two steps using 10% of the labelled patches (see Methods "Label data preparation"), in which all dead trees that are visible in NAIP images were manually digitised, resulting in more than 3000 dead tree polygons. We first evaluated the alignment of shapes (i.e., dead tree areas) between predictions and labels using the Intersection of Union (*IoU*) of each class (i.e., dead tree and background class) and the mean *IoU* of all classes (*mIoU*) was calculated as:

$$IoU = \frac{1}{n} \sum_{i=1}^{n} \frac{TP_i}{TP_i + FP_i + FN_i}$$
$$mIoU = \frac{1}{k} \sum_{i=1}^{k} IoU_i \qquad (4)$$

*TP* is the true positive value, meaning the number of pixels over overlapped areas, *FP* is the false positive value, meaning the number of pixels over overestimated areas, *FN* is the false negative value, meaning the number of pixels over underestimated areas, *k* is the number of classes.

We then evaluated the accuracy of the dead tree count following ref. 54 and calculated the count bias as:

$$bias = \frac{\sum_{i=1}^{n} (Y_i^{obs} - Y_i^{pred})}{\sum_{i=1}^{n} Y_i^{obs}} \times 100 \qquad (5)$$

The *IoU* for the dead tree class and background class were 0.53 and 1, respectively, which means that no false negative occurred and the dead tree crown size tended to be underestimated. The *mIoU* was 0.77, meaning a 77% accuracy when considering both background and dead tree classes. The ME (Eq. (3)) after bias correction was 0.56 dead tree ha-1 (Supplementary Fig. 1a). When comparing to manually interpreted dead trees from NAIP images, the overall underestimation bias (Eq. (5)) was 3.61% and the MAE (Eq. (1)) was 2.27 dead tree ha-1 (Supplementary Fig. 1b). Supplementary Fig. 1b shows that the cumulative error of dead tree count decreases with the increase of sample size, indicating that the dead tree count accuracy increases for larger areas. Given that fires have been one of the dominant abiotic causes of acute tree mortality in California, we calculated the bias (Eq. (5)) and

MAE (Eq. (1)) for dead trees found within and outside fire perimeters[46] to represent the accuracy of fire and non-fire-related tree mortality (Supplementary Fig. 2).

## Comparison to ground observations

The ground surveys of tree mortality used for the model evaluation were collected between 2016 and 2023 and consist of three types of tree mortality observations: (1) geo-locations of individual dead trees collected through opportunistic sampling, (2) geo-locations of all dead trees within each plot, and (3) the total number of dead trees within each plot. The evaluation against ground observations was conducted at both tree and plot levels. Standing dead trees observed in the field before 2020 may lose a large portion of the top branches or fall[42], which may not be seen in NAIP 2020 images and contribute to the omission errors. The collection time of NAIP images used for predictions and ground observations for validation need to be aligned, especially for areas with environmental conditions favouring fast decomposition and decay of dead trees. However, given the limited access to ground observations of tree mortality in California, we used all available data collected from 2016 to 2023 and reported the bias (Eq. (5)) and MAE of the count of dead trees per ha (Eq. (1)) by years whenever applicable.

The tree-level observations consist of point locations for 898 dead trees and crown polygons for 73 dead trees collected between 2016 and 2023 (Supplementary Table 1). The dead tree information in datasets named SMNB2016, SMNB2019, SMSB2020 (Supplementary Table 1), DS2021, and DS2023 (Supplementary Table 4), were collected through opportunistic sampling, and only the ones that were visible from remote sensing were sampled. DS2021 and DS2023 were only used to evaluate the predictions for NAIP 2022 to demonstrate how accurately the model trained using NAIP 2020 can map individual dead trees and estimate dead tree counts for NAIP images of other years. In MCVNB2018[77] and MCVNB2020, the geo-locations of all dead trees in a plot that were ≥1.35 m in height and ≥40 cm in DBH were recorded. MCVNB2020 also includes stressed trees ($n = (4)$) that have a high likelihood to die within a year of the sampling (possible imminent death, PIDs). The dead tree crown polygons collected in 2019 (SMNB2019) range between 38.45 and 587.22 m², while some polygons cover multiple crowns. The plot-level observations are the total count of dead trees for 88 plots, which consists of 13 plots sized between 0.93 and 3.35 ha visited in 2018[77] (Supplementary Fig. 3a) and 75 circular 0.1 ha-large plots visited in 2016[13] (Supplementary Fig. 3b–f). The largest plot (3.35 ha) visited in 2018 is a merge of three partially overlapped subplots to avoid double counting of dead trees within the overlaps. The count of dead trees at the plot level for 2018 was aggregated from point locations of 197 dead trees from MCVNB2018[77]. In the 2016 plot dataset[13,57] (DX2016), dead trees within a 0.1 ha-large plot that were ≥1.35 m in height and ≥cm DBH were recorded with information on species, DBH, and decay stages (Supplementary Fig. 3c). There were originally 98 plots visited, and 87 plots had coordinates information. Among the 87 plots, there were a total of 2921 dead trees recorded, of which 327 dead trees with DBH ≥40 cm were found in 75 plots. To be consistent with MCV2018, we only considered dead trees ≥40 cm for the plot-level evaluation of the count of dead trees per ha (Supplementary Fig. 3b). We also used DX2016 to calculate the percentage of overstory trees given the available information of DBH in this dataset. We first converted the DBH into three height classes (i.e., short: <15 m, medium: 15-30 m, tall: >30 m) based on species-specific allometric equations (Supplementary Table 7) following ref. 57. We then calculated the percentage of dead trees for each height and species group and considered trees classified as medium or tall class within the same species group as the overstory trees. A total of 12 dead trees were not included in this analysis given the missing information of species or the species types are not included in Supplementary Table 5. The bounding box of all in-situ datasets covered about 52,000 hectares in the southern Sierra Nevada (Supplementary Fig. 7), which suggests that our evaluation against ground observations is most representative of Sierra Nevada Range.

For the tree-level comparison, we generated a 6 m buffer for each predicted dead tree segment to mitigate the geo-referencing uncertainties of NAIP images. We then compared it to the point locations or polygons of dead trees in ground survey datasets. We identified the location of a predicted dead tree as correct when the predicted dead tree segment intersects with a dead tree point or more than 50% of a dead tree crown polygon recorded in the field. Based on Eq. (5), we identified the bias as the percentage of ground observations intersected with predicted dead tree crowns. The bias was first calculated for each dataset separately and then averaged by survey years.

For the plot-level comparison, we applied a 6 m buffer for each plot polygon (Supplementary Fig. 3c) and counted the centroids of predicted dead trees within each buffered plot and compared the number of dead trees over 40 m DBH in each plot. We calculated the bias (Eq. (5)) and MAE (Eq. (1)) for the comparison to plot observations in 2016 and 2018 separately.

## Characterisation of tree mortality from the individual dead tree map

To characterise tree mortality, we extracted structural metrics and mortality stages from the individual dead tree map. We define structural metrics as geometric properties of identified dead trees, such as the area and shape of dead tree crowns, count of dead trees, and percentage of dead trees. The brown- and grey-stage are associated with the spectrum of dead tree crowns. A brown-stage dead tree appears brownish or reddish on RGB images and normally implies recent death with leaves remaining on stems[48]. A grey-stage dead tree appears greyish or whitish and implies long-standing deadwood or snags[48]. In the following sections, we describe the retrievals of structural metrics and mortality stages from the individual dead tree map separately.

## Structural metrics of individual dead trees

We extracted geometric properties such as area (after filling hollows) and coordinates of centroid for each object on the individual dead tree map. The dead tree count map (Supplementary Fig. 9a) was then generated by aggregating the number of centroids of dead trees within 100 × 100 m and 240 × 240 m grids. The count maps are bias-corrected, meaning the ME of dead tree count per ha (Eq. (3)) calculated from the training set was added to the pixel values. The count map was then used to generate the percentage of tree mortality map (Supplementary Fig. 9e) at 240 m resolution by comparing it to the tree count map for 2011[43] at the pixel level.

We also created a map showing the percentage of dead canopy per ha by summing the area of dead tree crowns within 100 × 100 m grids (Supplementary Fig. 9c). The median of dead tree crown sizes within 100 × 100 m grid was used to represent the size of dead tree crowns per ha (Supplementary Fig. 9b). These datasets derived from the individual dead tree map can be produced at multiple resolutions depending on the interests of users. We used a 100 m resolution, which represents a hectare on the ground and closely responds to the expressed needs of forest managers.

## Colour space-based classification of mortality stages

We classified mortality stages such as brown- (recent mortality) and grey-stage (long-standing deadwood) based on the Hue-Saturation-Value (HSV) colour space of NAIP images following an approach adapted from ref. 48 We first applied a one-pixel inner buffer for dead tree segments to mitigate edge effects and calculated the mean of RGB values within each buffered segment. The mean RGB values were

converted into $H$, $S$, $V$, and used to calculate $X_{green}$, $X_{brown}$, $X_{grey}$, and $X_{background}$ as:

$$X_{green} = \begin{cases} \frac{C_g^{H_g} - 1}{C_g - 1}, & H_g \geq 0 \\ 0, & H_g < 0 \end{cases}$$

$$X_{brown} = \begin{cases} \frac{C_r^{H_r} - 1}{C_r - 1}, & H_r \geq 0 \\ 0, & H_r < 0 \end{cases}$$

$$X_{grey} = \frac{C_y^{1-S} - 1}{C_y - 1}$$

$$X_{background} = \frac{C_b^{1-V} - 1}{C_b - 1}$$

$$H_g = 1 - \frac{|H - 1/3|}{1/6}$$

$$H_r = \frac{1/2 - H}{1/6} - 2 \tag{6}$$

where the $C_g$, $C_r$, $C_y$, and $C_b$ are tunable parameters. We adopted the same settings used in ref. 48 (5, 5, 1e7, and 1e4), which were determined following training with NAIP data spanning from 2012 to 2019 for the entire US with an overall accuracy of 0.89-0.90.

The classification of mortality stages was then based on the maximum value among $X_{green}$, $X_{brown}$, $X_{grey}$, and $X_{background}$. If the maximum value is $X_{brown}$, the dead tree is classified as brown-stage mortality. After identifying the mortality stage of each dead tree, we calculated the percentage of brown-stage mortality against the number of all dead trees within 100 × 100 m grids (Supplementary Fig. 9d). To eliminate noise, we removed the pixels with only one brown-stage dead tree in this calculation.

We used the ADS survey data collected in 2020 (ADS2020) to validate the classification performance at the plot level. ADS2020 consists of more than 1000 hand-drawn polygons over areas where brown-stage trees were visually identified from very high-resolution images (0.25 to 0.6 m) as the flight campaign was cancelled. We laid the ADS2020 polygons over the count map of brown-stage dead trees (100 m resolution) and extracted the total number of brown-stage dead trees within each ADS polygon. To ensure at least one pixel from the count map within each ADS polygon, we discarded ADS polygons smaller than 1 pixel size (1 ha) in this analysis, resulting in 1,180 ADS polygons. False negatives (i.e., brown-stage misclassified as grey-stage) were only found in 116 polygons (9.8% of the total number of ADS polygons).

### Damage agents of tree mortality from expert knowledge

We used ADS datasets[10] and historical fire records[46,69] to attribute damage agents to dead trees falling inside ADS and fire polygons. In the ADS datasets, a maximum of three damage agents were logged to each polygon based on expert knowledge and ground surveys with a relatively high accuracy[21,22]. The ADS survey is conducted on a yearly basis and reports tree mortality that occurred between the previous and the current survey year. Therefore, we used the latest record of damage agents for overlapping areas between ADS polygons from different years (Eq. (7)). To complement ADS datasets, we used historical fire records[46,69] to identify fire-impacted areas. We followed Eq. (8) to determine the damage agents for overlapping areas between ADS and fire polygons. Hereafter, the post-processed ADS and fire datasets are referred to as damage agent polygons or damage agent datasets.

We found 85 different damage agent labels (Level 4 category; Supplementary Table 8) in the damage agent datasets, which are regrouped into 15 Level 3, nine Level 2, and three Level 1 categories. Level 2 category was used to visualise the spatial distribution of recent and primary damage agents (Supplementary Fig. 6a). Level 1 and Level 3 categories were used to analyse the composition of damage agents (Supplementary Fig. 6c and Supplementary Fig. 13). In total, only 13.6% of the total area covered by damage agent polygons were attributed to more than one Level 4 damage agent categories (Supplementary Fig. 13). By overlapping the damage agent polygons with the individual dead tree map, we also summarised the total number of dead trees by Level 1 and Level 3 damage agent categories (Supplementary Fig. 6c and Supplementary Table 9).

Drought-related mortality was mostly mapped as points in the ADS datasets. By overlapping these point layers with a species map[44], we identified that Oregon white oak, Canyon live oak/interior live oak, Blue oak, and Deciduous oak woodland were the dominant species that were prone to droughts. This is also in line with the findings in ADS reports. The number of dead trees in those areas was accounted as drought-caused mortality in Supplementary Fig. 6.

$$Agents_{ADS1 \cap ADS2} = \begin{cases} Agents_{ADS1}, & Year_{ADS1} \geq Year_{ADS2} \\ Agents_{ADS2}, & Year_{ADS1} < Year_{ADS2} \end{cases} \tag{7}$$

$$Agents_{ADS \cap Fire} = \begin{cases} (Agents_{ADS1 \cap ADS2}, Fire), & Year_{Fire} \geq Year_{ADS} \\ Agents_{ADS1 \cap ADS2}, & Year_{Fire} < Year_{ADS} \end{cases} \tag{8}$$

### Quantification of view angle-caused uncertainties

Severe geometric distortion caused by off-nadir view angles has been found on NAIP images near the edge of flight routes and in mountainous areas (Supplementary Fig. 9f). This has resulted in elongated ellipsoid-shaped dead tree crowns in the prediction (Supplementary Fig. 9f), which introduces uncertainties in predicted dead tree crown sizes and is considered one of the main reasons for a relatively low $IoU$ value for dead tree class in the model assessment. To quantify the severity of geometric distortion, we calculated an eccentricity for each predicted dead tree crown as:

$$e = \frac{c}{a} \tag{9}$$

where $e$ is eccentricity, $c$ is the distance between the focal points, $a$ is the length of the major axis.

The eccentricity is between 0 and 1 and is inversely proportional to the roundness of a dead tree crown. In practice, we calculated median eccentricity for dead tree crowns more than 50 pixels (18 m²) within 500 × 500 m grids. The median, mean, and standard deviation (std.) of eccentricities in California are 0.68, 0.68, and 0.12, respectively (Supplementary Fig. 9f). We recommend carefully using area-related maps (i.e., dead canopy area and median dead tree crown size) where eccentricity is larger than 0.8 (mean + std.).

### Reporting summary

Further information on research design is available in the Nature Portfolio Reporting Summary linked to this article.

## Data availability

NAIP images are freely available on Google Earth Engine (https://developers.google.com/earth-engine/datasets/catalogue/USDA_NAIP_DOQQ). The sources of all ancillary datasets are listed in Supplementary Table 6. Derived products, i.e., dead tree count per ha (100 m), median dead crown size per ha (100 m), percentage of dead canopy area per ha (100 m), percentage of brown-stage mortality per ha

(100 m), eccentricity map (500 m), and percentage of tree mortality (240 m), are freely accessible at ref. 78 Field observations DX2016 and MCVNB2018 are available from refs. 13,57 and ref. 77, respectively. Field observations (datasets SMNB2016, SMNB2019, MCVNB2020, SMSB2020, and DSSB2021) are available from A.D. and DS2023 is available from A.D. and Y.C.

## Code availability

The code for NAIP data downloading from GEE, the pre-trained EfficientUNet with the deep watershed algorithm for dead tree segmentation, and the post-segmentation processing code can be accessed at (https://doi.org/10.6084/m9.figshare.23723388). All codes used for this study were written in Python (3.9).

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

## Acknowledgements

This research was supported by the Villum Fonden (DRYTIP project, grant agreement no. 37465) and by the University of Copenhagen (PerformLCA project, UCPH Strategic plan 2023 Data+ Pool). S.H. and S.O. acknowledge the support from Villum Fonden through the research grant DeReEco (grant agreement no. 34306). S.O. further acknowledges the support from the Independent Research Fund Denmark through the research grant "Monitoring Changes in Big Satellite Data via Massively-Parallel Artificial Intelligence" (grant agreement no. 9131-00110B) and PerformLCA project (UCPH Strategic plan 2023 Data+ Pool). M.B. was

funded by the European Research Council (ERC) under the European Union's Horizon 2020 Research and Innovation Programme (grant agreement no. 947757 TOFDRY). M.B. and M.M. further acknowledge DFF Sapere Aude (grant agreement no. 9064–00049B). The authors acknowledge Ankit Kariryaa from the University of Copenhagen for his contributions to the development of the general deep learning frame-work for remote sensing data analysis. Y.C. warmly thanks the remote sensing group at the University of Copenhagen for all the insightful discussions, as well as Nicholas Ampersee, A.D., and Anne Hopkins Pfaff from U.S. Geological Survey for their support during the fieldwork in 2023 in the Sequoia National Park. Ground-based data collection was funded by the U.S. Geological Survey Ecosystems programme and Climate and Land Use Research and Development programme, with substantial additional support from the National Park Service. Any use of trade, firm, or product names is for descriptive purposes only and does not imply endorsement by the U.S. Government.

## Author contributions

Y.C. and S.H. designed the study, with input from M.B., L.R., A.D. and C.B. S.O. developed the deep learning code for dead tree segmentation with input from Y.C. Y.C. developed the code for post-segmentation feature extraction and mortality stage classification, supported by S.O. Y.C. prepared the training and test datasets, partially customised the image augmentation in the dead tree segmentation code, and trained the model, supported by S.O. A.D. and L.R. provided the ground observations. M.M. contributed to the model training. Y.C. conducted the analyses. Y.C., S.H., M.B., S.O., L.R. and A.D. conducted the interpretations. Y.C. and S.H. drafted the first manuscript. Y.C., S.H., M.B., A.V., A.D., W.V., L.R., C.B., S.O., M.M., S.S. and F.W. contributed to the final version of the manuscript. Y.C. designed and prepared the figures, with suggestions from S.H., M.B., A.V., S.O., L.R., C.B.

## Competing interests

The authors declare no competing interests.
