## [Peer Review File · Nature Communications]

Scattered tree death contributes to substantial forest loss in CaliforniaEditorial Note: Parts of this Peer Review File have been redacted as indicated to remove third-party material where no permission to publish could be obtained.

REVIEWER COMMENTS

Reviewer #1 (Remarks to the Author):

Review of NCOMMS-23-30672

Widespread isolated tree death contributes to substantial forest loss in California

Yan Cheng, Stefan Oehmcke, Martin Brandt, Lisa Rosenthal, Adrian Das, Anton Vrieling, Sassan Saatchi, Fabien Wagner, Maurice Mugabowindekwe, Wim Verbrugge, Claus Beier, Stephanie Horion

This study develops new methodology for identifying and classifying condition of individual trees from NAIP imagery, and applies the method to assess tree mortality across the entire state of California in 2020. This study highlights the value of fine grain information across broad spatial extents and simultaneously pushes the cutting edge of tree detection/classification while advancing our fundamental understanding of California forest ecology. This is a clever approach to focus on dead individual tree detection instead of trying to go for both dead and live tree detection right off the bat. A key result was the very common detection of tree mortality in isolated patches of 3 or fewer trees, and even of only a single tree. This phenomenon is of course known to occur, but understanding its full extent and the degree to which it contributes to forest demography in California can't be understood without an approach such as this one. It was also very well-written, and was therefore a pleasure to read.

I have a couple of more major points, and a few minor ones, all of which I would think would be readily addressable in a revision.

Major points:

L445: For opportunistic sampling of field reference trees, it seems like the IoU metric is impossible to calculate because you can't determine false positives, right? You have to have a full census of trees within a known area (e.g., a plot) in order to say that a NAIP/deep learning detected tree matches to a field reference tree. With opportunistic sampling, a lack of a field reference tree co-located with a NAIP/deep learning detected tree could be because there is no field reference tree there or just because it wasn't sampled. How is this handled? How can you adequately assess model skill against field reference trees without a full census?

There are a few places where being more explicit about "grain/resolution" and "extent" rather than just talking about "scale" will help clarify the messaging.

I would be very interested to read a discussion point about how readily this approach might scale across years. Have the authors tried running inference on prior years' NAIP data, even if it's just on a few tiles, to see how readily temporal trends in tree mortality are detected?

This approach also classifies dead trees versus background, and I'd be interested to read any kind of discussion point about how good the approach might be at detecting live trees too. For both this point and the one above (about temporal trends), I would understand if the authors declare this out of scope to do any further analysis, but I do think it's where the heads of forest ecologists and managers will be after reading your work so might be worth commenting on!

Minor Points:

L66: It might be worth adding something about "...are therefore needed across broad extents." to help clarify your message that both fine grain and broad extend tree mortality surveys are key.

L88: What is meant by "large-scale mortality" here?

L89: Is the non-fire related mortality also of the "large-scale mortality" variety here too? Please specify, if so.

L88-89: Can you specify whether the bias was an over- or underestimate for the fire and non-fire induced tree mortality? I suspect underestimates for all of these given how it is worded, but it is a little ambiguous.

L111: "gathering half of all detected dead trees". Maybe "accounting for" instead of "gathering"? "Accounting for half of all detected dead trees"?

L124-125: "Fires occur throughout California and indiscriminately affect forest types." The other statements around this are in past tense, making them more tethered to your specific findings. E.g., "The majority of inset-related tree mortality was found in conifer-dominated woodlands". I think with the line about fires, you are trying to say that fire-induced tree mortality was found in all of the 16 main forest type groups and wasn't especially dominant in any of them-- is that right? As written, the sentence feels detached from your actual tree mortality findings and has the appearance of saying that, in general, "fires occur throughout California" (true) and "fires indiscriminately affect forest types" (I wouldn't agree with this, since fire is a very different phenomenon depending on forest type. That is, its behavior and effects strongly depend on forest type). I think this is just a point in need of a little clarification in that your detected fire-induced tree mortality was not especially prominent across forest types though!

L155-157: Would you mind adding the percentage of total dead trees represented by solo trees (with the 19.5 million number) and the total number of dead trees represented by small groups of dead trees (with the 60% number)? This would help the reader paint a fuller picture. I realize the reader can do the math with the total number you've presented (91.4 million), but it'd be helpful to have both measures in the same place! (That is, the number of trees in each category, and the fraction of the total represented by each category). Thanks!

L205: You explain the %brown and the dead tree count relevance, but what is the value of the median crown size? This could also relate to recency of tree mortality? Or forest type?

L218: I have to admit I'm having a hard time understanding the RGB color space plot. I can see that it is a slick way to try to convey three axes of forest structure measurements in a single color, but I'm not finding it intuitive. I'll defer to the authors, but have you considered breaking this out into a separate single color gradient for each of the three metrics?

L249-250: I think it'd be worth swapping in "resolution" for the first instance of "scale" and "extent" for the second instance of "scale", which would help clarify this excellent point you are making in these lines.

L346: I'm not understanding the math in this section. Each NAIP patch was 256 pixels x 256 pixels and pixels have a 0.6 meter spatial resolution. That's $256 \times 0.6 \times 256 \times 0.6$ meters squared == 23,600 meters squared. You say you sampled 714 of those patches and $714 \times 23,600$ == 16,850,000 square meters (16,850 hectares). Is that right? Where does the 109,670 number come in? Finding ~27,000 dead trees in only 109,670 square meters seems like that's not quite right (way too high). What am I missing?

L364: Did you tune these numbers? Or are these hyper parameter values known to be good?

L372-376: Can you expand on this? Why is it good or needed to "predict the presence of low energy before higher energy levels can be predicted"?

Reviewer #2 (Remarks to the Author):

The presented manuscripts address a highly pertinent subject within the framework of the contemporary global challenge of accelerating forest mortality rates. The authors have adeptly intertwined cutting-edge remote sensing techniques with geospatial driver analysis, resulting in an insightful exploration. This study offers a promising pathway towards comprehending the intricate dynamics underlying forest health in these changing times. The manuscript is overall clearly

structured and nicely written. Find below some major suggestions / questions followed by some detailed comments.

Major comments:

1) I think the research gap is not that precisely formulated. The authors only refer to Landsat, while not naming state-of-the-art satellite missions that are more suitable (Sentinel, Planet-Labs, Worldview, Pleiades). There are several publications using these missions that successfully mapped tree mortality (some of which are cited here). In my opinion, the main gap addressed here is that is about quantifying tree mortality at the individual level and not the area level (see a more detailed comment below).

2) In the introduction, the authors themselves state that tree mortality is a complex phenomenon that can depend on multiple drivers. Their attribution to individual events in the methods section does not seem to reflect this complexity (if I correctly understand their approach). For instance, did the fire occur after or before the death of trees? Are the insect infestations a result of drought-weakened trees? It's a chicken-and-egg-problem, but putting such simplified numbers is very dangerous without being very clear. Institutions will use such numbers for their management and if you do not provide context, they may do wrong decisions. There is also no explanation on how these percentages are related to each other (l.122, what's 100 %?). I also have a specific question about the drought: How do you know among all the different species if the drought played a role in the mortality event? Each species reacts differently to droughts, given their differences biophysical properties (rooting depth, isohydry, canopy height, etc...) and their biotic and abiotic environment. Our definition of droughts are usually based on statistics (e.g. an statistically determined extreme event through time), while this does not necessarily relate to the mortality event. Thus; I suggest to have a more detailed analysis or nuanced presentation/interpretation of these results.

3) In the discussion you mention that you could apply this procedure on a yearly basis. Why did you just apply it to one year then? Using several years you could easily test how the brown-stage relates to mortality time. Moreover, it would be very interesting to check the consistency of your predictions across years. A multitemporal analysis could enable to more robustly link the forest mortality dynamics to biotic/abiotic dynamics (e.g. recent droughts).

4) I do not fully understand why the authors use a semantic segmentation for individual tree detection. State-of-the-art algorithms would use a instance segmentation directly. From a computer vision perspective, a semantic segmentation is the wrong solution for this task, while an instance segmentation would be much more efficient. Here, the authors counter the limitation of a semantic segmentation by applying a watershed algorithm, which seems to work fine, but appears to be an overcomplicated and less efficient workaround.

Detailed comments:

Title: I think isolated tree death sounds a bit confusing. Maybe "scattered" is a better term?

l.44 maybe "compound events" could be a useful term here.

l.64 I agree, for the problem mentioned here (resolution) Landsat is not ideal suitable. Yet, there are several studies that use Sentinel-2 data for mapping tree mortality and forest decline, which is globally available and comes with a higher spectral quality and resolution. Also Planet-data might be an option. Consider to be more precise here about the state of the art and the research gap: Mapping tree mortality/forest decline/disturbance is possible with large-scale satellite data (Sentinel/Landsat) but is restricted to indirect estimates via vegetation index anomalies (e.g. Senf et al. 2021, Nature sustainability, Furniss et al. RSE), indirectly through biochemical properties (e.g. Zarco-Tejada et al. 2019, RSE) or by mapping the area coverage of dead trees (Campbell et al. 2020, RSE; Schiefer et al. 2023, ISPRS Open). Yet, none of these approaches enable to count individuals, and this is what you aim at here.

l.64. Again, I would be more precise here. What you address here is not the extent (implies an area), but the counts of dead trees.

l.68. Consider to be more specific and use machine learning- or deep learning-based computer vision / pattern recognition

l.73 The uncertainty of the 'predicted' density.

l.130 Do you really mean wind? Why? Drying out trees via increased evaporation pressure? Or do you mean storms?

l.132 Indeed, it relies on multiple agents and I would mention this actually before presenting your analysis that these results have to be interpreted with great care.

Fig. 1: The circles occlude most of the raster areas. I suggest to put the circles next to the annotation outside the raster. Else the raster is not of any value.

I.151 Why did you not use a state-of-the-art instance segmentation which can do this right away? (see major comment above).

I.187 Where did you show this correlation? This is very species depended, or? I would expect large differences among coniferous and deciduous trees when it comes to 'holding' on their leaves and respective crown color. Not accounting for such systematic effects may lead to systematic biases (e.g. through species distribution). I would at least check if you can observe such differences between deciduous/conifers and probably apply to different models.

I.209 This is where I would be careful: If there are different brown_color vs. time_since_mortality-relationships among species, the interpretation here may partly result from the species distribution. I guess a separation into coniferous / deciduous trees should be possible to check and potentially account for such effects. Otherwise, this should be clearly discussed in my opinion.

I.270 Can you report a rough fraction based on field data how much of the dead trees are commonly assumed to be over-story trees?

I.525 Have you considered to validate this approach?

I.534 Could be an interesting resource to test the brown-stage-age-relationship across species (see comment above).

I.276 Maybe consider to have a designated conclusion section? In any case, I would more prominently highlight your main findings here. There were a lot of innovative analysis here. So whats new (and the cardinal take home message) from your study (beyond the method) that advances the understanding of tree mortality in California?

References:

- Senf, C., & Seidl, R. (2021). Mapping the forest disturbance regimes of Europe. *Nature Sustainability*, 4(1), 63-70.
- Zarco-Tejada et al. (2019). Chlorophyll content estimation in an open-canopy conifer forest with Sentinel-2A and hyperspectral imagery in the context of forest decline. *Remote sensing of environment*, 223, 320-335.
- Campbell et al. (2020). A multi-sensor, multi-scale approach to mapping tree mortality in woodland ecosystems. *Remote Sensing of Environment*, 245, 111853.
- Schiefer et al. (2023). UAV-based reference data for the prediction of fractional cover of standing deadwood from Sentinel time series. *ISPRS Open Journal of Photogrammetry and Remote Sensing*, 8, 100034.
- Furniss, T. J., Kane, V. R., Larson, A. J., & Lutz, J. A. (2020). Detecting tree mortality with Landsat-derived spectral indices: Improving ecological accuracy by examining uncertainty. *Remote Sensing of Environment*, 237, 111497.

Response to Reviewers

Note: To facilitate the reading of our answers to comments, changes of text are indicated in green. Unless it is stated, line numbers refer to the track changes version for easy referral. Figures in the response letter are named Fig. R1, Fig. R2, etc..

+++++

Reviewer #1:

This study develops new methodology for identifying and classifying condition of individual trees from NAIP imagery, and applies the method to assess tree mortality across the entire state of California in 2020. This study highlights the value of fine grain information across broad spatial extents and simultaneously pushes the cutting edge of tree detection/classification while advancing our fundamental understanding of California forest ecology. This is a clever approach to focus on dead individual tree detection instead of trying to go for both dead and live tree detection right off the bat. A key result was the very common detection of tree mortality in isolated patches of 3 or fewer trees, and even of only a single tree. This phenomenon is of course known to occur, but understanding its full extent and the degree to which it contributes to forest demography in California can't be understood without an approach such as this one. It was also very well-written, and was therefore a pleasure to read.

I have a couple of more major points, and a few minor ones, all of which I would think would be readily addressable in a revision.

Major points:

(R1C1) L445: For opportunistic sampling of field reference trees, it seems like the IoU metric is impossible to calculate because you can't determine false positives, right? You have to have a full census of trees within a known area (e.g., a plot) in order to say that a NAIP/deep learning detected tree matches to a field reference tree. With opportunistic sampling, a lack of a field reference tree co-located with a NAIP/deep learning detected tree could be because there is no field reference tree there or just because it wasn't sampled. How is this handled? How can you adequately assess model skill against field reference trees without a full census?

***Authors:** The model predictions (dead tree maps) were evaluated by comparing to two different datasets: 1) field survey datasets which consist of three types of observations of dead trees, i.e. geo-locations of individual dead trees collected through opportunistic sampling, geo-locations of all dead trees within plots, and the total numbers of dead trees within plots, and 2) manual labels (i.e. the test set) where all dead trees visible in the NAIP images within plots sized 256 × 256 pixel were manually digitised (see *Methods* “Label data preparation”, L409 - 415). We further clarified the field observations used for model evaluation in the section titled “Comparison to ground observations” as follows (L510 - 513): “The ground surveys of tree mortality used for the model evaluation were collected*

between 2016 and 2023 and consist of three types of tree mortality observations: 1) geolocations of individual dead trees collected through opportunistic sampling, 2) geolocations of all dead trees within each plot, and 3) the total number of dead trees within each plot.”

(m)IoU was used to evaluate the model performance against manual labels and not for the field observations. (m)IoU demonstrates the classification performance of dead trees and background objects (e.g., alive trees, soil, and others). Therefore the (m)IoU can unravel the false positive at the pixel level, i.e. background pixels classified as dead tree pixels. Aside from (m)IOU, we also calculated the Mean Absolute Error (MAE) for the count of dead trees per ha (Equation (1)) when comparing to the manual labels. We further elaborated on the corresponding lines in the section titled “Model performance” as follows (L482 - 486 and L494 - 495): “The model performance was evaluated in two steps using 10% of the labelled patches (see Methods “Label data preparation”) in which all dead trees that are visible in NAIP images were manually digitised, resulting in more than 3,000 dead tree polygons. We first evaluated the alignment of shapes (i.e. dead tree areas) between predictions and labels using the Intersection of Union (IoU) of each class (i.e. dead tree and background class) and the mean of all classes (mIoU) was calculated as:...” and “We then evaluated the accuracy of dead tree count following ref.⁵⁴ and calculated the count bias as:...”

The comparison to field survey data was to assess the underestimation bias which was quantified using different metrics than (m)IOU. This comparison was conducted at both tree and plot level. At the plot level, we compared the predicted count of dead trees to the total number of dead trees recorded in the field survey for each plot and calculated MAE of count of dead trees per ha (Equation (1)) and Mean Error (ME) of count of dead trees per ha (Equation (3)). This can be considered as a comparison to full census data given that all dead trees (>1.35 m in height or >40 cm in DBH) within a plot were recorded in the field survey. The tree-level comparison was quantified through the bias (Equation (5)), which demonstrated the overall underestimation bias of the count of dead trees. We elaborated the comparison to field observations in the section titled “Comparison to ground observations” as follows (L513 - 514): “The evaluation against ground observations was conducted at both tree and plot level. ... we used all available data collected from 2016 to 2023 and reported the bias (Equation (5)) and MAE of count of dead trees per ha (Equation (1)) by years whenever applicable.”

We acknowledge that the field observations used in this study are concentrated in Southern Sierra Nevada due to the limited data availability and accessibility. Nevertheless, given the similar species composition and topography across the Sierra Nevada Range (i.e. the ecoregion 5 in Extended Data Fig. 2), the evaluation against these field observations can represent the model performance for most of the Sierra Nevada area, which covers about 5.2 million hectares (18.7% of California) and dominant by conifer forests. In addition, the test set (i.e. 10% of total labels) is located across California covering diverse species groups and elevation ranges (mean±std.: 1,539±916m). The evaluation against the test set is therefore an indicator of model performance across the entire study area. To allow for making a more comprehensive comparison to field observations in the future, we stress the need for standardised data collection protocols and data sharing. We elaborated on this in the Discussion section as follows (L316 - 321):

“First, the ground observations are geographically limited to the Sierra Nevada Range and species-wise dominant by conifers; therefore, they may not represent the entire California. Despite that, the model was also evaluated against manually digitised labels across California that cover diverse species groups and elevation ranges (mean \pm std.: 1,539 \pm 916 m; Supplementary Fig. 2). Nevertheless, standardised data collection and data sharing protocols for tree mortality are necessary to facilitate more comprehensive evaluation against ground observations.”

(R1C2) There are a few places where being more explicit about "grain/resolution" and "extent" rather than just talking about "scale" will help clarify the messaging.

Authors: *We thank the reviewer for pointing out to be more explicit and cautious when using different terms in relation to spatial extent and resolution. We have double-checked the relevant lines. Aside from updating the lines pointed out by the reviewer in the detailed comments, we also updated the following lines:*

“Near-surface remote sensing technologies, such as UAV- and airborne-based optical and LiDAR, have recently been explored as an alternative method for fine-resolution_forest health monitoring...” (L55 - 56)

“As fire spread is strongly related to spatial heterogeneity of fuel density and fuel flammability at fine resolutions...” (L307 - 308)

“The fine-resolution mapping may also support improved carbon accounting as well as fire risk and behaviour modelling.” (L340 - 342)

(R1C3) I would be very interested to read a discussion point about how readily this approach might scale across years. Have the authors tried running inference on prior years' NAIP data, even if it's just on a few tiles, to see how readily temporal trends in tree mortality are detected?

Authors: *We followed the reviewer’s suggestion. We applied the model to map dead trees from three additional years of NAIP images, i.e. 2016, 2018, and 2022 (the latest available NAIP images on Google Earth Engine) for a spatial subset of the study area. We showcased the predictions of individual dead trees and the changes in the number of dead trees per ha for several mortality scenarios (e.g., recent mortality and succession “alive tree - brown stage - grey stage”) in Fig. 4 (L264 - 270): “Fig. 4: Tree mortality mapping from multi-year NAIP images (2016, 2018, 2020, and 2022) for a spatial subset of the study area. a, NAIP images from 2016 - 2020. b, predictions of individual dead trees. c, count of detected dead trees per ha. d-g, detailed views of multi-year NAIP images and detected dead trees: d, most recent mortality from NAIP 2022; e, increased tree mortality from 2016 to 2022; f, decreased number of dead trees detected due to the manual removal of dead trees or the natural decomposition / falling of dead trees; g, spreading of tree mortality from the north to the south. The locations of these four detailed views are indicated with black bounding boxes in Panel c for 2022.”*

We evaluated the accuracy of multi-year mapping by comparing to the field observations collected in or before the mapping year. For example, the predictions for 2018 were evaluated by comparing them to the dead trees recorded in the field surveys of 2018 and 2016. We then calculated the bias (Equation (5)) when comparing to field observations in

2018 and the average bias for 2018 and 2016. The evaluation results are summarised in the Extended Data Table 2. The additional datasets used to evaluate predictions for 2022 are described in the Methods section as follows (L522 - 528): “The tree-level observations consist of point locations for 898 dead trees and crown polygons for 73 dead trees collected between 2016 and 2023 (Extended Data Table 1). The dead tree information in dataset SMNB2016, SMNB2019, SMSB2020 (Extended Data Table 1), DS2021, and DS2023 (Extended Data Table 2) ... DS2021 and DS2023 were only used to evaluate the predictions for NAIP 2022 to demonstrate how accurately the model trained using NAIP 2020 can map individual dead trees and estimate dead tree counts for NAIP images of other years.”

We added a section titled “Multi-year tree mortality mapping” to demonstrate the feasibility of multi-year tree mortality mapping from NAIP images using the model trained on one-year data (L244 - 262): “We directly applied the model trained using NAIP images in 2020 to the NAIP images from 2016, 2018, and 2022 over a spatial subset of the study area. The model was able to detect the most recent mortality from NAIP 2022 (Fig. 4d), the gradual increase in mortality from 2016 to 2022 (Fig. 4e), and the decreased number of dead trees due to manual removal, natural falling or decomposition (Fig. 4f). Fig. 4d and 4e illustrate some of the inconsistencies in between-year image distortions and georeferencing of NAIP imagery, which introduces challenges for comparing changes at the individual tree level. However, this challenge is reduced when aggregating to count of dead trees per ha (Fig. 4c), which effectively shows temporal trends of tree mortality over the test area. The spatial-temporal patterns of tree mortality (Fig. 4d-g) provide detailed information for future studies in drivers and help improve the early warning of tree mortality.”

By comparing to field surveys of dead trees or snags from 2016 to 2023 (Methods), we found an underestimation bias between 16.7% and 53.1% (Extended Data Table 2). The underestimation biases for 2016, 2018, and 2022 are higher than for 2020, which could be a result of different geometrical and spectral characteristics of the 2016, 2018, and 2022 images with respect to the 2020 images that were the basis of the training samples. Further addition of training samples for the different years may improve the accuracy of multi-year mapping. The highest bias was found in predictions for 2022 when comparing to field observations before 2021, which could be partially explained by the fire disturbances that occurred in 2021 over the field sites.”

We also added a discussion point as follows (L285 - 289): “Applying the model to map individual dead trees for multi-year NAIP images over a test area, we showed that a clear trend can be identified in the count of dead trees per ha, in spite of inconsistent performances across years. Given the availability of NAIP images, this demonstrates the potential to map tree mortality rate for the contiguous United States on a biennial basis while accurately accounting for scattered dead trees.”

Lastly, in the last paragraph in the Introduction section, where we introduce the objectives or main activities of this study, we added corresponding lines as follows (L80 - 83): “Lastly, we presented an example of multi-year tree mortality mapping by applying the model trained for 2020 directly to NAIP images acquired in adjacent years over a subset of the study area and evaluated the accuracy against field observations.”

(R1C4) This approach also classifies dead trees versus background, and I'd be interested to read any kind of discussion point about how good the approach might be at detecting live trees too. For both this point and the one above (about temporal trends), I would understand if the authors declare this out of scope to do any further analysis, but I do think it's where the heads of forest ecologists and managers will be after reading your work so might be worth commenting on!

***Authors:** We agree with the reviewer that live tree mapping is important in understanding forest ecosystems and change within. We have made our framework easy to customise and train a new model that detects living trees with appropriately annotated data, i.e. the segmentations of living trees. For example, Mugabowindekwe et al. (2022) tested a simpler version of UNet architecture for mapping living trees on NAIP images in California, with convincing results (see Extended Data Fig. 8 in Mugabowindekwe et al. (2022)). This is indeed encouraging and we do have ongoing work on this topic within the team of authors, but as the reviewer states, out of the scope of this study. We have added a discussion point as follows (L297 - 299): "In addition, similar deep learning frameworks have been used to map live trees from high-resolution satellite and aerial images³⁹⁻⁴¹, which in combination with the tree mortality maps enables the study of tree regeneration after forest degradation."*

Reference:

[1] Mugabowindekwe, M. et al. Nation-wide mapping of tree-level aboveground carbon stocks in Rwanda. Nat. Clim. Change 1–7 (2022) doi:10.1038/s41558-022-01544-w.

Minor Points:

L66: It might be worth adding something about "...are therefore needed across broad extents." to help clarify your message that both fine grain and broad extend tree mortality surveys are key.

***Authors:** We thank the reviewer's suggestion. This line has been updated accordingly (L64 - 65): "Assessments at the tree level based on finer resolution data are therefore needed across broad extents."*

L88: What is meant by "large-scale mortality" here?

***Authors:** Most fire-related tree mortality or burned trees appear to be in clusters, which is in contrast to scattered die-offs. To avoid confusion, it has been removed from the previous text (L91): "The overall underestimation bias and MAE for fire-related mortality..."*

L89: Is the non-fire-related morality also of the "large-scale mortality" variety here too? Please specify, if so.

Authors: *Non-fire-related mortality can also be either large- or small-scales. To avoid confusion, the “large-scale mortality” has been removed from the previous text (L87): “The overall underestimation bias and MAE for fire-related mortality...”*

L88-89: Can you specify whether the bias was an over- or underestimate for the fire and non-fire induced tree mortality? I suspect underestimates for all of these given how it is worded, but it is a little ambiguous.

Authors: *We thank the reviewer for the suggestion. This line has been updated accordingly (L87): “The overall underestimation bias and MAE for fire-related mortality...”*

L111: "gathering half of all detected dead trees". Maybe "accounting for" instead of "gathering"? "Accounting for half of all detected dead trees"?

Authors: *We thank the reviewer for the suggestion. This line has been updated accordingly (L113 - 114): “...California mixed conifer appeared to be the most affected, accounting for half of all detected dead trees...”*

L124-125: "Fires occur throughout California and indiscriminately affect forest types." The other statements around this are in past tense, making them more tethered to your specific findings. E.g., "The majority of insect-related tree mortality was found in conifer-dominated woodlands". I think with the line about fires, you are trying to say that fire-induced tree mortality was found in all of the 16 main forest type groups and wasn't especially dominant in any of them-- is that right? As written, the sentence feels detached from your actual tree mortality findings and has the appearance of saying that, in general, "fires occur throughout California" (true) and "fires indiscriminately affect forest types" (I wouldn't agree with this, since fire is a very different phenomenon depending on forest type. That is, its behavior and effects strongly depend on forest type). I think this is just a point in need of a little clarification in that your detected fire-induced tree mortality was not especially prominent across forest types though!

Authors: *We fully agree with the reviewer that the behaviours and effects of fires are highly related to forest structures and species compositions (Agee, 1996). We originally wanted to emphasise that the fires have occurred in different forest type groups (Extended Data Fig. 6). To avoid confusion, we have removed “indiscriminately” from the original text (L132 - 133): “Fires occurred throughout California and affected all forest types.”*

Reference:

[1] Agee, J. K. (1996). *The influence of forest structure on fire behavior. In Proceedings of the 17th annual forest vegetation management conference (pp. 52-68).*

L155-157: Would you mind adding the percentage of total dead trees represented by solo trees (with the 19.5 million number) and the total number of dead trees represented by small groups of dead trees (with the 60% number)? This would help the reader paint a fuller picture. I realize the reader can do the math with the total number you've presented (91.4 million), but

it'd be helpful to have both measures in the same place! (That is, the number of trees in each category, and the fraction of the total represented by each category). Thanks!

Authors: *We thank the reviewer for the suggestion. We have updated the text and the figure as suggested (L158, Fig. 2). The percentage of total dead trees represented by solo trees is 28.4% (based on the total number of dead trees prior bias correction). As we mentioned in the main text, the total number of dead trees at the state level (~91.4 million) is an aggregation from the number of dead trees per ha over the entire study area. The number of dead trees per ha has been bias-corrected following Igel et al. (2023; see L470 - 472). The reason for bias correction is as follows. The best-performed model was selected based on the Mean Absolute Error (MAE) of count of dead trees per patch on the validation set (Methods, see L463 - 465). However, the main task of the model is segmentation, for which the training does not minimise MAE directly, so the sum of residuals cannot be expected to be zero. Therefore there will be a systematic error for the count of dead trees. This systematic error can accumulate into a greater bias with the increase of prediction area. Therefore, if the interest is aggregated values such as the total number of dead trees in California, it is important to correct the bias as a post-processing step. When analysing the dead tree density at 30 m resolution and comparing to Landsat-derived forest loss data, we are interested in the subset-level values instead of the aggregated values. The 30 m-resolution dead tree density map was therefore directly derived by counting the number of dead tree segments within each 30 × 30 m grid. The total number of dead trees used to calculate the percentages in this figure is simply the sum of the number of dead trees per 30 × 30 m grid over the study area, which results in a total number of dead trees of ~68.6 million. We further elaborated on it in the Methods section as follows (L478 - 480): “For tree-level or subset-level analysis, for example, when comparing predicted dead tree density at 30 m resolution to Landsat-derived forest loss data, the count of dead trees for each 30 × 30 m grid was directly derived from individual dead tree segmentations without bias correction.”*

Reference:

[1] Igel, C. & Oehmcke, S. Remember to correct the bias when using deep learning for regression! *KI - Künstliche Intelligenz* 37, 33–40 (2023).

L205: You explain the %brown and the dead tree count relevance, but what is the value of the median crown size? This could also relate to recency of tree mortality? Or forest type?

Authors: *As rightfully pointed out by the reviewer, the median crown size of dead trees can be an indicator of the decay stages (Hayden et al. 1995). When the dead tree is classified as a brown-stage dead tree, i.e. still retaining its foliage, the median crown size in combination with species information can be used as a proxy of tree sizes/diameters (Jucker et al. 2017). More data and analysis are still needed to quantify the relationship between crown size and the decay stages as it can vary across species groups and age groups. However we believe that, in addition to the dead tree count, the median crown size is a relevant structural characteristic of dead trees that can be used to understand the decay stages in combination with other information such as species. We further elaborated on why this is an important characteristic of a dead tree in the text as follows (L185 - 187): “The median of dead tree crown size per ha can be used to understand the*

decay stages and tree sizes/diameters in combination with ancillary information such as species⁴⁹.

[REDACTED]

Reference:

[1] Hayden, J., J. Kerley, D. Carr, T. Kendi, and J. Hallarn. 1995 Field manual for establishing and measuring permanent sample plots. Queen's Printer for Ontario, Toronto, ON

[2] Jucker, T. et al. Allometric equations for integrating remote sensing imagery into forest monitoring programmes. Global Change Biology 23, 177–190 (2016).

L218: I have to admit I'm having a hard time understanding the RGB color space plot. I can see that it is a slick way to try to convey three axes of forest structure measurements in a single color, but I'm not finding it intuitive. I'll defer to the authors, but have you considered breaking this out into a separate single color gradient for each of the three metrics?

***Authors:** We thank the reviewer for the suggestion on map visualization. The single-color map can be found in Extended Data Fig. 7. We decided to keep Fig. 3 as we believe Extended Data Fig. 7 might be too large to be placed in the main text. The updated figure caption is as follows (L227 - 242): "Fig. 3: Characterisation of tree mortality at the landscape scale based on structural metrics and mortality stages of individual dead trees. ... Separate maps for each of the three metrics are provided in Extended Data Fig. 7. Panels a-f are examples of six generic types of tree mortality, as characterised based on dead tree count ..."*

L249-250: I think it'd be worth swapping in "resolution" for the first instance of "scale" and "extent" for the second instance of "scale", which would help clarify this excellent point you are making in these lines.

***Authors:** We thank the reviewer for the suggestion. We have updated the text accordingly (L295 - 297): "..., this enables holistic profiles of existing mortality at fine resolutions, directly supporting the monitoring of forest health at large extents and the understanding of the interaction between climate and biotic stressors⁵⁴."*

L346: I'm not understanding the math in this section. Each NAIP patch was 256 pixels x 256 pixels and pixels have a 0.6 meter spatial resolution. That's 256*0.6 * 256*0.6 meters squared

== 23,600 meters squared. You say you sampled 714 of those patches and $714 * 23,600 == 1685000$ square meters (1685 hectares). Is that right? Where does the 109670 number come in? Finding ~27,000 dead trees in only 109,670 square meters seems like that's not quite right (way too high). What am I missing?

Authors: *We thank the reviewer for pointing this out. We have revised the number to 1,685 hectares (L409): “We sampled 714 patches with a size of 256×256 pixels from the NAIP images (~1,685 hectares)...”*

L364: Did you tune these numbers? Or are these hyper parameter values known to be good?

Authors: *We tested different combinations for alpha and beta, i.e. (0.3, 0.7), (0.4, 0.6), (0.6, 0.4). For gamma, we used 2 as it is recommended by the paper (Abraham et al. 2019). When gamma is larger than 1, it becomes a focal loss, meaning “difficult” samples with high individual loss are weighted higher than samples with low individual loss. This has proven to perform accurately with imbalanced datasets as what we have in this study, i.e. more background pixels than dead tree pixels. When alpha is smaller than beta, the model will focus on minimising false positive predictions (Abraham et al. 2019), which results in relatively conservative tree mortality maps. Li et al. (2023) and Mugabowindekwe et al. (2023) have used unequal alpha and beta in the Tversky loss function for mapping trees from aerial photos. Among all tested combinations of alpha, beta, and gamma values, we found that (0.4, 0.6, 2) achieved the highest predictive performance on the validation set, i.e. lowest in MAE of count of dead trees per ha. We then used this combination to train the final model. We elaborated on this in the corresponding lines as follows (L426- 431): “The Focal Tversky loss function⁶⁹ together with the Adam optimiser was used for model parameter optimisation. We set alpha, beta, and gamma values in the Focal Tversky loss function as 0.4, 0.6, and 2. When the gamma value is greater than one, the focal loss part in the loss function is activated, which has proven a good performance with imbalanced datasets⁶⁹. When the alpha value is larger than the beta value, the model focuses on minimizing false positive predictions⁶⁹, which results in relatively conservative predictions for dead trees.”*

Reference:

[1] Abraham, N., & Khan, N. M. (2019, April). A novel focal tversky loss function with improved attention u-net for lesion segmentation. In 2019 IEEE 16th international symposium on biomedical imaging (ISBI 2019) (pp. 683-687). IEEE.

[2] Li, S. et al. Deep learning enables image-based tree counting, crown segmentation and height prediction at national scale. PNAS Nexus pgad076 (2023) doi:10.1093/pnasnexus/pgad076.

[3] Mugabowindekwe, M. et al. Nation-wide mapping of tree-level aboveground carbon stocks in Rwanda. Nat. Clim. Change 1–7 (2022) doi:10.1038/s41558-022-01544-w.

L372-376: Can you expand on this? Why is it good or needed to "predict the presence of low energy before higher energy levels can be predicted"?

Authors: *We have now elaborated on the benefits of an ordinal classification as follows (L442 - 448): “This adaptation assures that 1) the energy levels represent the distance from the border, which results in an intuitive and meaningful label of the predicted segments, 2) no unrealistic holes remain within the predicted dead tree segments, 3) tree object boundaries can potentially be sharpened or refined during postprocessing by removing lower energy levels, which would be a more informed action than simple shrinking, 4) consistency is required for easy separation of closely grouped instance (e.g., a change in the direction of energy level gradient serves as an indicator of a new instance).”*

Reviewer #2:

The presented manuscripts address a highly pertinent subject within the framework of the contemporary global challenge of accelerating forest mortality rates. The authors have adeptly intertwined cutting-edge remote sensing techniques with geospatial driver analysis, resulting in an insightful exploration. This study offers a promising pathway towards comprehending the intricate dynamics underlying forest health in these changing times. The manuscript is overall clearly structured and nicely written. Find below some major suggestions / questions followed by some detailed comments.

Major comments:

(R2C1) I think the research gap is not that precisely formulated. The authors only refer to Landsat, while not naming state-of-the-art satellite missions that are more suitable (Sentinel, Planet-Labs, Worldview, Pleiades). There are several publications using these missions that successfully mapped tree mortality (some of which are cited here). In my opinion, the main gap addressed here is that is about quantifying tree mortality at the individual level and not the area level (see a more detailed comment below).

***Authors:** We agree with the reviewer that the research gap in the previous manuscript was not adequately described. We have revised the corresponding text in the introduction with more references on satellite image-based forest degradation mapping as follows (L58 - 63): “Systematic assessment of forest degradation at regional to landscape is possible with satellite images such as those from Sentinel-2 and Landsat²⁶⁻³³. Finer resolution images such as PlanetScope, WorldView, and Pleiades have also been tested to map tree mortality³⁴⁻³⁶. Restricted to area-level estimates via vegetation index anomalies^{26-29,33-36}, biochemical properties³⁰, or by mapping deadwood fractions at the pixel level^{31,32}, these approaches can not readily count individuals and likely miss scatter dead trees³⁷.”*

(R2C2) In the introduction, the authors themselves state that tree mortality is a complex phenomenon that can depend on multiple drivers. Their attribution to individual events in the methods section does not seem to reflect this complexity (if I correctly understand their approach). For instance, did the fire occur after or before the death of trees? Are the insect infestations a result of drought-weakened trees? Its a chicken-and-egg-problem, but putting such simplified numbers is very dangerous without being very clear. Institutions will use such numbers for their management and if you do not provide context, they may do wrong decisions. There is also no explanation on how these percentages are related to each other (l.122, what's 100 %?). I also have a specific question about the drought: How do you know among all the different species if the drought played a role in the mortality event? Each species reacts differently to droughts, given their differences biophysical properties (rooting depth, isohdry, canopy height, etc...) and their biotic and abiotic environment. Our definition of droughts are usually based on statistics (e.g. an statistically determined extreme event through time), while this does not necessarily relate to the mortality event. Thus; I suggest to have a more detailed analysis or nuanced presentation/interpretation of these results.

Authors: We thank the reviewer for the comment and agree that the current version of the manuscript did not adequately reflect the fact that tree mortality is often the result of multiple causal agents. Here, the ADS database is our main source of information which are hand-drawn polygons indicating areas with tree mortality and contains information on up to three different causal agents observed from the plane or on the ground for each polygon (logged in the database as primary, secondary, and tertiary). Previously, we focused on the primary causal agents. To answer the reviewer's comment, we analysed all three levels of causal agents in the database. To complement the ADS database we also used annual fire perimeters from 2012 to 2020 produced by CAL FIRE and fuel disturbance data from LANDFIRE to identify fire-impacted areas. The ADS and fire datasets together are referred to as damage agent polygons or damage agent datasets hereafter. The result showed only about 13.6% of the total area in damage agent datasets was attributed to multiple causal agents (e.g., a combination of abiotic and biotic damage agents or multiple species of pests and diseases; see Supplementary Fig. 5). To the best of our knowledge and considering the information provided by the damage agent datasets, we regard this number as the current best reflection of the phenomenon for the State of California. However, we also stay critical and acknowledge the possibility of non-recorded secondary causal agents. We now revised the Extended Data Fig. 6 to include information on multiple damage agents and revised the figure caption as follows (L954 - 964): “Fig. 6: Damage agent attribution based on ADS database and fire perimeters from 2012-2020. a, spatial distribution of the most recent and dominant damage agents (Level 2 category). b, showcases of tree mortality related to six dominant damage agents (Level 2 category): b1, Insects (e.g., Bark Beetles), b2, Fire, b3, Drought, b4, Diseases, b5, Wild animal (e.g., Wild boars), b6, Human activity (e.g., Herbicides). Each panel consists of two items. The top shows the true colour NAIP image, and the bottom shows the dead tree predictions. The black lines on the edge of each image represent 30 × 30 m grids. The geolocation of each sample area is indicated on the left-hand map. c, percentage of total number of dead trees associated with each type of damage agent (Level 1 and Level 3 category). Group A consists of two categories of multiple-damage-agent: Human Activities and Abiotic Agents (0.0011%) and Human Activities and Biotic Agents (0.00014%). The detailed information for Group B, C, and D is listed in Supplementary Table 7.”

We updated the results about damage agents in the main text as follows (L122 - 136): “Given the interactions between damage agents, tree mortality may be attributed to multiple damage agents^{12–14,22}, such as a combination of biotic and abiotic agents or multiple species of pests or diseases. In particular, the prolonged droughts in California since 2012 have been suggested as one of the underlying triggers of bark-beetle-related tree mortality¹⁴. By overlapping the individual dead tree map with damage agent surveys¹⁰ and fire perimeters⁴² from 2012 to 2020 (referred to as damage agent database hereafter, Methods), we assigned potential damage agents for 67.7% of the detected dead trees. As shown in the Extended Data Fig. 6, biotic agents such as bark beetles (42.5%), cankers (1.5%), wood borers (0.89%), and combinations of bark beetles and wood borers (0.89%) were the dominant damage agent of tree mortality in California, accounting for 45.8% of total dead trees detected, followed by fires (27.2%), drought (16.5%), and a combination of bark beetles and fires (7.1%). The majority of biotic agent-related tree mortality was found in conifer-dominated woodlands. Fires occurred throughout California and affected all forest types. Single-agent drought-related mortality was mostly observed on oak woodlands in the foothills of central California. Wild animals, such as wild boar, and human

activities, such as herbicides, were found to have a minor impact on tree mortality (1.2% and 0.33%; Extended Data Fig. 6).”

We updated the methods for damage agents as follows (L623 - 641): “We used ADS datasets¹⁰ and historical fire records^{42,60} to attribute damage agents to dead trees falling inside ADS and fire polygons. In the ADS datasets, a maximum of three damage agents (excluding fires) were logged to each polygon based on expert knowledge and ground surveys with a relatively high accuracy^{21,22}. The ADS survey is conducted on a yearly basis and reports tree mortality that occurred between the previous and the current survey year. Therefore, we used the latest record of damage agents for overlapping areas between ADS polygons from different years (Equation (7)). To complement ADS datasets, we used historical fire records^{42,60} to identify fire-impacted areas. We followed Equation (8) to determine the damage agents for overlapping areas between ADS and fire polygons. Hereafter, the post-processed ADS and fire datasets are referred as damage agent polygons or damage agent datasets.

We found 85 different damage agents (Level 4 category; Supplementary Table 6) in the damage agent datasets, which are regrouped into 15 Level 3, nine Level 2, and three Level 1 categories. Level 2 category was used to visualise the spatial distribution of recent and primary damage agents (Extended Data Fig. 6a). Level 1 and Level 3 categories were used to analyse the composition of damage agents (Extended Data Fig. 6c and Supplementary Fig. 5). In total, only 13.6% of the total area covered by damage agent polygons were attributed to more than one Level 4 damage agent categories (Supplementary Fig. 5). By overlapping damage agent polygons with the individual dead tree map, we also summarised the total number of dead trees by Level 1 and Level 3 damage agent categories (Extended Data Fig. 6c and Supplementary Table 7).”

We updated the corresponding lines in the Discussion section as follows (L289 - 297): “We also explored the dominant damage agents of tree mortality in California from 2012 to 2020 by overlapping dead tree maps with ancillary datasets of damage agents documented through aerial and ground surveys. We found that bark beetles and fires are related to nearly 70% of the dead trees mapped in this study. Despite uncertainties in the ancillary datasets, the attribution of damage agents at the tree level provides spatially detailed information supporting future studies on spatial patterns of damage agents. Together with the temporal dimension, this enables holistic profiles of existing mortality at fine spatial resolution, directly supporting the monitoring of forest health for large extents and the understanding of the interaction between climate and biotic stressors⁵⁴.”

We also acknowledge the uncertainties of damage agent attributions coming along with the limitations in ADS datasets. We added a discussion point as follows (L331 - 337): “Fourth, tree mortality can be a result of compound events and interactions between multiple damage agents^{1,12–14,22,54}, which has proven to be challenging to document exhaustively through aerial or ground surveys, including the data used in this study (ADS)²². In particular, it is difficult to attribute drought especially when compounded with other damage agents such as bark beetles or fires. Nevertheless, we provide the first tree-level analysis on damage agents at the state scale, which can serve as first-hand material to understand the spatial and temporal patterns of tree mortality related to different damage agents.”

In the previous L122, the first percentage refers to the percentage of areas with dead trees detected, and the second percentage is the percentage of the predicted number of dead trees. Therefore, these two percentages are independent to each other. To avoid confusion, we simplified this line as follows (L127): “...we assigned potential damage agents for 67.7% of the detected dead trees.”

Reference:

[1] Coleman, T. W. et al. Accuracy of aerial detection surveys for mapping insect and disease disturbances in the United States. For. Ecol. Manag. 430, 321–336 (2018).

[2] Department of Forestry and Fire Protection. CAL FIRE Fire Perimeters through 2021. <https://frap.fire.ca.gov/mapping/gis-data/> (2021).

[3] LANDFIRE Program. Fuel Disturbance (FDIST). <https://landfire.gov/fdist.php>.

(R2C3) In the discussion you mention that you could apply this procedure on a yearly basis. Why did you just apply it to one year then? Using several years you could easily test how the brown-stage relates to mortality time. Moreover, it would be very interesting to check the consistency of your predictions across years. A multitemporal analysis could enable to more robustly link the forest mortality dynamics to biotic/abiotic dynamics (e.g. recent droughts).

***Authors:** We followed the reviewer’s suggestion. We applied the model to map dead trees from three additional years of NAIP images, i.e. 2016, 2018, and 2022 (the latest available NAIP images on Google Earth Engine) for a spatial subset of the study area. We showcased the predictions of individual dead trees and the changes in the number of dead trees per ha for several mortality scenarios (e.g., recent mortality and succession “alive tree - brown stage - grey stage”) in Fig. 4 (L264 - 270): “Fig. 4: Tree mortality mapping from multi-year NAIP images (2016, 2018, 2020, and 2022) for a spatial subset of the study area. a, NAIP images from 2016 - 2020. b, predictions of individual dead trees. c, count of detected dead trees per ha. d-g, detailed views of multi-year NAIP images and detected dead trees: d, most recent mortality from NAIP 2022; e, increased tree mortality from 2016 to 2022; f, decreased number of dead trees detected due to the manual removal of dead trees or the natural decomposition / falling of dead trees; g, spreading of tree mortality from the north to the south. The locations of these four detailed views are indicated with black bounding boxes in Panel c for 2022.”*

We evaluated the accuracy of multi-year mapping by comparing to the field observations collected in or before the mapping year. For example, the predictions for 2018 were evaluated by comparing them to the dead trees recorded in the field surveys of 2018 and 2016. We then calculated the bias (Equation (5)) when comparing to field observations in 2018 and the average bias for 2018 and 2016. The evaluation results are summarised in the Extended Data Table 2. The additional datasets used to evaluate predictions for 2022 are described in the Methods section as follows (L522 - 528): “The tree-level observations consist of point locations for 898 dead trees and crown polygons for 73 dead trees collected between 2016 and 2023 (Extended Data Table 1). The dead tree information in dataset SMNB2016, SMNB2019, SMSB2020 (Extended Data Table 1), DS2021, and DS2023 (Extended Data Table 2) ... DS2021 and DS2023 were only used to evaluate the

predictions for NAIP 2022 to demonstrate how accurately the model trained using NAIP 2020 can map individual dead trees and estimate dead tree counts for NAIP images of other years.”

We added a section titled “Multi-year tree mortality mapping” to demonstrate the feasibility of multi-year tree mortality mapping from NAIP images using the model trained on one-year data (L244 - 262): “We directly applied the model trained using NAIP images in 2020 to the NAIP images from 2016, 2018, and 2022 over a spatial subset of the study area. The model was able to detect the most recent mortality from NAIP 2022 (Fig. 4d), the gradual increase in mortality from 2016 to 2022 (Fig. 4e), and the decreased number of dead trees due to manual removal, natural falling or decomposition (Fig. 4f). Fig. 4d and 4e illustrate some of the inconsistencies in between-year image distortions and georeferencing of NAIP imagery, which introduces challenges for comparing changes at the individual tree level. However, this challenge is reduced when aggregating to count of dead trees per ha (Fig. 4c), which effectively shows temporal trends of tree mortality over the test area. The spatial-temporal patterns of tree mortality (Fig. 4d-g) provide detailed information for future studies in drivers and help improve the early warning of tree mortality.

By comparing to field surveys of dead trees or snags from 2016 to 2023 (Methods), we found an underestimation bias between 16.7% and 53.1% (Extended Data Table 2). The underestimation biases for 2016, 2018, and 2022 are higher than for 2020, which could be a result of different geometrical and spectral characteristics of the 2016, 2018, and 2022 images with respect to the 2020 images that were the basis of the training samples. Further addition of training samples for the different years may improve the accuracy of multi-year mapping. The highest bias was found in predictions for 2022 when comparing to field observations before 2021, which could be partially explained by the fire disturbances that occurred in 2021 over the field sites.”

We also added a discussion point as follows (L285 - 289): “Applying the model to map individual dead trees for multi-year NAIP images over a test area, we showed that a clear trend can be identified in the count of dead trees per ha, in spite of inconsistent performances across years. Given the availability of NAIP images, this demonstrates the potential to map tree mortality rate for the contiguous United States on a biennial basis.”

Lastly, in the last paragraph in the Introduction section, where we introduce the objectives or main activities of this study, we added corresponding lines as follows (L80 - 83): “Lastly, we presented an example of multi-year tree mortality mapping by applying the model trained for 2020 directly to NAIP images acquired in adjacent years over a subset of the study area and evaluated the accuracy against field observations.”

We agree with the reviewer that mapping tree mortality for multiple and continuous years can provide detailed information to understand the relationship between the brown stage and the timing of death. Indeed, as we showcase in Fig. 4d-g and Fig. R2 in this response letter, the majority of the conifer (such as pine) and deciduous trees (such as oak) turned from brown to grey stage within two years. However, ground observations are needed to conduct a true validation of the mortality stage classification and to study the relationship between mortality stages and the timing of death. We hope to work together with the

collaborators from USGS to establish a database to further study this. Despite that, the method we used to detect the tree mortality stage was trained using NAIP data from 2012 - 2019 for all 48 States in the contiguous US and has reported an accuracy of 0.89 - 0.9 (Monahan et al. 2022). In the main text, we further emphasised that the mortality stage classification in this study is a direct indication of a dead tree with or without dried or decoloured foliage (L192 - 195 and L217), but more datasets are needed to quantify the relationship between the mortality stage and the timing of death for different species.

Reference:

[1] Monahan, W. B. et al. A spectral three-dimensional color space model of tree crown health. PLOS ONE 17, e0272360 (2022)

(R2C4) I do not fully understand why the authors use a semantic segmentation for individual tree detection. State-of-the-art algorithms would use a instance segmentation directly. From a computer vision perspective, a semantic segmentation is the wrong solution for this task, while an instance segmentation would be much more efficient. Here, the authors counter the limitation of a semantic segmentation by applying a watershed algorithm, which seems to work fine, but appears to be an overcomplicated and less efficient workaround.

Authors: We thank the reviewer for the question. Using semantic segmentation for individual tree detection in aerial images is a well-established practice, as evidenced by recent work (Tucker et al. 2023, Li et al. 2023, Mugabowindekwe et al. 2023). It is a valid approach given the context of the problem.

For example, the tree detection work of Brandt et al. (2020) assumes that trees are mostly stand-alone and separating them is not required, so instance segmentation would introduce unnecessary complexity. When counting is required and trees are grouped closer, density-based adaptations of semantic segmentation have been applied before (Li et al. 2023). More recent work (Tucker et al. 2023 and Mugabowindekwe et al. 2023) on instance segmentation from semantic segmentation for tree detection uses an adapted weighting of the loss function to achieve a clear separation. The common idea with these methods is that simple adaptations specific to the task are preferred over applying more complex methods (e.g., such as RCNN or YOLO variants).

We follow in that spirit with our approach by utilising an energy-based approach, which effectively is simply learning multiple semantic segmentation outputs simultaneously. The subsequent application of the watershed algorithm is efficient and uses a well-established computer vision method (Beucher 1991 and Beucher et al. 1976). We chose not to make this a model comparison paper because our primary focus is on achieving accurate results and their in-depth analysis, aligning with our research objectives. While general instance segmentation is a viable alternative, our approach offers unique advantages for this specific task without significantly increasing the model complexity, as is reflected by the performance metrics. The specific advantages of the watershed algorithm are as follows: 1) the energy levels represent the distance from the border, which results in an intuitive and meaningful label of the predicted segments, 2) tree object boundaries can potentially be sharpened or refined during postprocessing by removing lower energy levels, which would be a more informed action than simple shrinking, 3) easy separation of closely

grouped instance (e.g., a change in the direction of energy level gradient serves as an indicator of a new instance).

Reference:

[1] Tucker, C., Brandt, M., Hiernaux, P. et al. Sub-continental-scale carbon stocks of individual trees in African drylands. *Nature* 615, 80–86 (2023).

<https://doi.org/10.1038/s41586-022-05653-6>

[2] Mugabowindekwe, M. et al. Nation-wide mapping of tree-level aboveground carbon stocks in Rwanda. *Nat. Clim. Change* 1–7 (2022) doi:10.1038/s41558-022-01544-w.

[3] Li, S. et al. Deep learning enables image-based tree counting, crown segmentation and height prediction at national scale. *PNAS Nexus* pgad076 (2023) doi:10.1093/pnasnexus/pgad076.

[4] Brandt, M. et al. An unexpectedly large count of trees in the West African Sahara and Sahel. *Nature* 587, 78–82 (2020).

[5] S. Beucher. The watershed transformation applied to image segmentation. *Scanning Microscopy International*, Suppl:6(1):299–314, 1991. 1, 2

[6] S. Beucher and C. Lantuejoul. Use of watersheds in contour detection. *Proc. Int. Workshop Image Processing, Real-Time Edge and Motion Detection/Estimation*, 1976.

Detailed comments:

Title: I think isolated tree death sounds a bit confusing. Maybe "scattered" is a better term?

Authors: *We thank the reviewer for the suggestion. We have revised the title to "Widespread scattered tree death contributes to substantial forest loss in California".*

I.44 maybe "compound events" could be a useful term here.

Authors: *We thank the reviewer for the suggestion. We revised the line as follows (L41 - 41): "A large portion of tree deaths was a result of compound events, such as droughts, bark beetles, and wildfires."*

I.64 I agree, for the problem mentioned here (resolution) Landsat is not ideal suitable. Yet, there are several studies that use Sentinel-2 data for mapping tree mortality and forest decline, which is globally available and comes with a higher spectral quality and resolution. Also Planet-data might be an option. Consider to be more precise here about the state of the art and the research gap: Mapping tree mortality/forest decline/disturbance is possible with large-scale satellite data (Sentinel/Landsat) but is restricted to indirect estimates via vegetation index anomalies (e.g. Senf et al. 2021, *Nature sustainability*, Furniss et al. RSE), indirectly through biochemical properties (e.g. Zarco-Tejada et al. 2019, RSE) or by mapping the area

coverage of dead trees (Campbell et al. 2020, RSE; Schiefer et al. 2023, ISPRS Open). Yet, none of these approaches enable to count individuals, and this is what you aim at here.

Authors: *We thank the reviewer for the suggested paraphrase. We revised the corresponding text in the introduction with additional references on satellite image-based forest degradation mapping as follows (L58 - 63): “Systematic assessment of forest degradation at regional to landscape is possible with satellite images such as those from Sentinel-2 and Landsat²⁶⁻³³. Finer resolution images such as PlanetScope, WorldView, and Pleiades have also been tested to map tree mortality³⁴⁻³⁶. Restricted to area-level estimates via vegetation index anomalies^{26-29,33-36}, biochemical properties³⁰, or by mapping deadwood fractions at the pixel level^{31,32}, these approaches can not readily count individuals and likely miss scatter dead trees³⁷.”*

I.64. Again, I would be more precise here. What you address here is not the extent (implies an area), but the counts of dead trees.

Authors: *We agree with the reviewer that the main research gap this article addressed is deriving a more straightforward estimate of the count of dead trees through mapping of tree mortality at the individual tree level. We have revised the corresponding line as follows (L63 - 64): “Consequently, the actual counts of dead trees and the contribution of scattered dead trees over large areas remain unknown.”*

I.68. Consider to be more specific and use machine learning- or deep learning-based computer vision / pattern recognition

Authors: *We thank the reviewer for the suggestion. We revised the corresponding line as follows (L67 - 69): “Advanced computer vision algorithms create the opportunity to effectively apply such imagery for fine-resolution and large-area mapping of tree mortality.”*

I.73 The uncertainty of the 'predicted' density.

Authors: *We thank the reviewer for pointing this out. We revised the corresponding line as follows (L72 - 73): “The uncertainty of predicted dead tree density was assessed with ground observations at the tree and plot levels.”*

I.130 Do you really mean wind? Why? Drying out trees via increased evaporation pressure? Or do you mean storms?

Authors: *We used damage agent information from the ADS database, where wind-tornado/hurricane (referred to as “wind” in the article) is one of the abiotic damage agents (Supplementary Table 6).*

I.132 Indeed, it relies on multiple agents and I would mention this actually before presenting your analysis that these results have to be interpreted with great care.

Authors: *We thank the reviewer for the suggestion. We updated the corresponding lines as follows (L122 - 127): “Given the interactions between damage agents, tree mortality may be attributed to multiple damage agents^{12–14,22}, such as a combination of biotic and abiotic agents or multiple species of pests or diseases. In particular, the prolonged droughts in California since 2012 have been suggested as one of the underlying triggers of bark-beetle-related tree mortality¹⁴. By overlapping the individual dead tree map with damage agent surveys¹⁰ and fire perimeters⁴² from 2012 to 2020 (referred to as damage agent database hereafter, Methods), we assigned potential damage agents for 67.7% of the detected dead trees. ...”*

Fig. 1: The circles occlude most of the raster areas. I suggest to put the circles next to the annotation outside the raster. Else the raster is not of any value.

Authors: *We thank the reviewer for pointing it out. The figure has been updated accordingly (L138).*

I.151 Why did you not use a state-of-the-art instance segmentation which can do this right away? (see major comment above).

Authors: *We thank the reviewer for the question. Using semantic segmentation for individual tree detection in aerial images is a well-established practice, as evidenced by recent work (Tucker et al. 2023, Li et al. 2023, Mugabowindekwe et al. 2023). It is a valid approach given the context of the problem.*

For example, the tree detection work of Brandt et al. (2020) assumes that trees are mostly stand-alone and separating them is not required, so instance segmentation would introduce unnecessary complexity. When counting is required and trees are grouped closer, density-based adaptations of semantic segmentation have been applied before (Li et al. 2023). More recent work (Tucker et al. 2023 and Mugabowindekwe et al. 2023) on instance segmentation from semantic segmentation for tree detection uses an adapted weighting of the loss function to achieve a clear separation. The common idea with these methods is that simple adaptations specific to the task are preferred over applying more complex methods (e.g., such as RCNN or YOLO variants).

We follow in that spirit with our approach by utilising an energy-based approach, which effectively is simply learning multiple semantic segmentation outputs simultaneously. The subsequent application of the watershed algorithm is efficient and uses a well-established computer vision method (Beucher 1991 and Beucher et al. 1976). We chose not to make this a model comparison paper because our primary focus is on achieving accurate results and their in-depth analysis, aligning with our research objectives. While general instance segmentation is a viable alternative, our approach offers unique advantages for this specific task without significantly increasing the model complexity, as is reflected by the performance metrics. The specific advantages of the watershed algorithm are as follows: 1) the energy levels represent the distance from the border, which results in an intuitive

and meaningful label of the predicted segments, 2) tree object boundaries can potentially be sharpened or refined during postprocessing by removing lower energy levels, which would be a more informed action than simple shrinking, 3) easy separation of closely grouped instance (e.g., a change in the direction of energy level gradient serves as an indicator of a new instance).

Reference:

[1] Tucker, C., Brandt, M., Hiernaux, P. et al. Sub-continental-scale carbon stocks of individual trees in African drylands. *Nature* 615, 80–86 (2023).

<https://doi.org/10.1038/s41586-022-05653-6>

[2] Mugabowindekwe, M. et al. Nation-wide mapping of tree-level aboveground carbon stocks in Rwanda. *Nat. Clim. Change* 1–7 (2022) doi:10.1038/s41558-022-01544-w.

[3] Li, S. et al. Deep learning enables image-based tree counting, crown segmentation and height prediction at national scale. *PNAS Nexus* pgad076 (2023) doi:10.1093/pnasnexus/pgad076.

[4] Brandt, M. et al. An unexpectedly large count of trees in the West African Sahara and Sahel. *Nature* 587, 78–82 (2020).

[5] S. Beucher. The watershed transformation applied to image segmentation. *Scanning Microscopy International*, Suppl:6(1):299–314, 1991. 1, 2

[6] S. Beucher and C. Lantuejoul. Use of watersheds in contour detection. *Proc. Int. Workshop Image Processing, Real-Time Edge and Motion Detection/Estimation*, 1976.

I.187 Where did you show this correlation? This is very species depended, or? I would expect large differences among coniferous and deciduous trees when it comes to 'holding' on their leaves and respective crown color. Not accounting for such systematic effects may lead to systematic biases (e.g. through species distribution). I would at least check if you can observe such differences between decideous/conifers and probably apply to different models.

Authors: We agree with the reviewer that how long the dead tree holds the foliage can be species-dependent (Hicke et al. 2012). By “the timing of death”, we actually meant two generic timing categories (mortality stages) depending on the existence of dried or decoloured foliage: 1) trees died recently, and still have foliage; 2) trees have been dead for a relatively long period compared to the same species at the brown stage, and lost all foliage. For the same species, age group, and damage agent, the existence of foliage (brown stage) implies more recent death as compared to dead trees with no foliage (grey stage). In addition, the existence of foliage has a strong correlation to the flammability of a dead tree, which regulates fire behaviours (Stephens et al. 2018). In short, the mortality stages directly indicate the existence of foliage, which is related to the flammability and the recency of mortality, but needs to be interpreted carefully along with ancillary datasets. In future studies, more data and in-depth analyses are needed to precisely quantify the relationship between dead crown colours and the timing of death across different species, age, and damage agent groups. We updated the corresponding lines as follows:

“The mortality stages such as brown-stage and grey-stage are determined by the dead canopy colour and imply the existence of foliage which can be a proxy for the recency of mortality when comparing within the same species group.” (L192 - 194)

“Differentiating between brown- and grey-stage allows for crude approximations of recent mortality from single-year NAIP images, along with species and damage agent information.” (L197 - 198)

“...first, newly developed mortality (stressed / dead trees with dried or decoloured foliage), denoted in greenish colours (Fig. 3c), refers to areas with high percentages of brown-stage mortality...” (L217 - 218)

Reference:

[1] Hicke, J. A., et al. *Effects of bark beetle-caused tree mortality on wildfire. Forest Ecology and Management* 271, 81–90 (2012).

[2] Stephens, S. L. et al. *Drought, Tree Mortality, and Wildfire in Forests Adapted to Frequent Fire. BioScience* 68, 77–88 (2018).

I.209 This is where I would be careful: If there are different brown_color vs. time_since_mortality-relationships among species, the interpretation here may partly result from the species distribution. I guess a separation into coniferous / deciduous trees should be possible to check and potentially account for such effects. Otherwise, this should be clearly discussed in my opinion.

Authors: *We agree with the reviewer that how long the dead tree holds the foliage can be species-dependent (Hicke et al. 2012). By “the timing of death”, we actually meant two generic timing categories (mortality stages) depending on the existence of dried or decoloured foliage: 1) trees died recently, and still have foliage; 2) trees have been dead for a relatively long period compared to the same species at the brown stage, and lost all foliage. For the same species, age group, and damage agent, the existence of foliage (brown stage) implies more recent death as compared to dead trees with no foliage (grey stage). In addition, the existence of foliage has a strong correlation to the flammability of a dead tree, which regulates fire behaviours (Stephens et al. 2018). In short, the mortality stages directly indicate the existence of foliage, which is related to the flammability and the recency of mortality, but needs to be interpreted carefully along with ancillary datasets. In future studies, more data and in-depth analyses are needed to precisely quantify the relationship between dead crown colours and the timing of death across different species, age, and damage agent groups. We updated the corresponding lines as follows:*

“The mortality stages such as brown-stage and grey-stage are determined by the dead canopy colour and imply the existence of foliage which can be a proxy for the recency of mortality when comparing within the same species group.” (L192 - 194)

“Differentiating between brown- and grey-stage allows for crude approximations of recent mortality from single-year NAIP images, along with species and damage agent information.” (L197 - 198)

“...first, newly developed mortality (stressed / dead trees with dried or decoloured foliage), denoted in greenish colours (Fig. 3c), refers to areas with high percentages of brown-stage mortality...” (L217 - 218)

Reference:

[1] Hicke, J. A., et al. Effects of bark beetle-caused tree mortality on wildfire. *Forest Ecology and Management* 271, 81–90 (2012).

[2] Stephens, S. L. et al. Drought, Tree Mortality, and Wildfire in Forests Adapted to Frequent Fire. *BioScience* 68, 77–88 (2018).

I.270 Can you report a rough fraction based on field data how much of the dead trees are commonly assumed to be over-story trees?

Authors: We do not have direct information on the percentage of overstory dead trees from field data used in this study. Tree heights can be an indicator of tree size and sometimes imply whether a tree is an overstory tree depending on the forest structure and species composition. We therefore analysed the tree heights for dead trees observed in a field survey in 2016 (DX2016; see Methods section for more information on this dataset). We first converted DBH classes into three height classes (i.e. short: >15 m, medium: 15-30 m, tall: >30 m) based on species-specific allometric equations (Supplementary Table 5) suggested by Stephenson et al. (2020). We then calculated the percentage of dead trees for each height and species group. The breakdown of tree height classes across species groups (Supplementary Table 3) indicates that 41.6% ($\pm 19.7\%$) of dead trees belong to medium or tall classes. Here, we only considered species groups with at least two height classes (i.e. small and medium) in Supplementary Table 5, as tree height classes lose correlation to overstory trees if there is only one height class. We updated the corresponding lines in the Discussion section as follows (L324 - 326): “As Supplementary Table 3 shows, there were only 41.6% ($\pm 19.7\%$) of dead trees belonging to medium or tall classes (≥ 15 m in height estimated from DBH using species-specific allometry equations⁵⁹).”

We also updated the Methods section accordingly as follows (L542 - 547): “We also used DX2016 to calculate the percentage of overstory trees given the available information of DBH in this dataset. We first converted the DBH into three height classes (i.e. short: >15 m, medium: 15-30 m, tall: >30 m) based on species-specific allometric equations (Supplementary Table 5) following ref.⁵⁹. We then calculated the percentage of dead trees for each height and species group and considered trees classified as medium or tall class within the same species group as the overstory trees.”

Reference:

[1] Stephenson, N. L. & Das, A. J. Height-related changes in forest composition explain increasing tree mortality with height during an extreme drought. *Nat. Commun.* 11, 3402 (2020).

I.525 Have you considered to validate this approach?

Authors: We thank the reviewer for the suggestions. We implemented an additional analysis to validate brown-stage classification at the plot level. We elaborated this analysis in the Methods section as follows (L613 - 621): “We used the ADS survey data collected

in 2020 (ADS2020) to validate the classification performance at the plot level. ADS2020 consists of more than 1,000 hand-drawn polygons over areas where brown stage trees were visually identified from very high-resolution images (0.25 to 0.6 m) as the flight campaign was cancelled. We laid the ADS2020 polygons over the count map of brown-stage dead trees (100 m resolution) and extracted the total number of brown-stage dead trees within each ADS polygon. To ensure at least one pixel from the count map within each ADS polygon, we discarded ADS polygons smaller than 1 pixel size (1 ha) in this analysis, resulting in 1,180 ADS polygons. False negatives (i.e. brown-stage misclassified as grey-stage) were only found in 116 polygons (9.8% of the total number of ADS polygons).”

However, to perform an in-depth tree-level evaluation, additional ground observations of mortality stages are needed. We hope to work together with the collaborators from USGS to establish a database and validate the classification of mortality stages and to understand the mortality stage and the timing of death. Nevertheless, the model for mortality stage classification used in this study was trained using NAIP data from 2012 - 2019 for all 48 States in the contiguous US and has reported an accuracy of 0.89 - 0.90 (see L603 - 605).

Reference:

[1] Monahan, W. B. et al. A spectral three-dimensional color space model of tree crown health. PLOS ONE 17, e0272360 (2022)

I.534 Could be an interesting resource to test the brown-stage-age-relationship across species (see comment above).

Authors: *We followed the reviewer’s suggestion and looked into the brown-stage-mortality-timing-relationship based on the ADS survey in 2020 and NAIP images from 2018 and 2020. The ADS data were used to locate the most recent mortality. The NAIP images were used to identify the colour of dead crowns over ADS areas. Through visual inspections, we found that for most species, it takes less than two years to turn from brown to grey stage (Fig. R2). However, additional ground observations are needed to quantify the relationship between mortality stages and the timing of death, we therefore decided not to include this analysis in the manuscript. We hope to work together with the collaborators from USGS to establish a database to further study this.*

Fig. R2: Examples of the changes from brown to grey stage across species.

I.276 Maybe consider to have a designated conclusion section? In any case, I would more prominently highlight your main findings here. There were a lot of innovative analysis here. So what's new (and the cardinal take home message) from your study (beyond the method) that advances the understanding of tree mortality in California?

Authors: *We thank the reviewer for the suggestion. We have extended the dedicated paragraphs in the Discussion section to elaborate on the main novelties and findings as follows (L285 - 299): "...Applying the model to map individual dead trees for multi-year NAIP images over a test area, we showed that a clear trend can be identified in the count of dead trees per ha, in spite of inconsistent performances across years. Given the availability of NAIP images, this demonstrates the potential to map tree mortality for the contiguous United States on a biennial basis while accurately accounting for scattered dead trees. We also explored the dominant damage agents of tree mortality in California from 2012 to 2020 by overlapping dead tree maps with ancillary datasets of damage agents documented through aerial and ground surveys. We found that bark beetles and fires are related to nearly 70% of the dead trees mapped in this study. Despite uncertainties in the ancillary datasets²², the attribution of damage agents at the tree level provides spatially detailed information supporting future studies on spatial patterns of damage agents. Together with the temporal dimension, this enables holistic profiles of existing mortality at fine spatial resolution, directly supporting the monitoring of forest health for large extents and the understanding of the interaction between climate and biotic stressors⁵⁴. In addition, similar deep learning frameworks have been used to map live trees from high-resolution satellite and aerial images³⁹⁻⁴¹, which in combination with the tree mortality maps enables the study of tree regeneration after forest degradation. ..."*

References:

- Senf, C., & Seidl, R. (2021). Mapping the forest disturbance regimes of Europe. *Nature Sustainability*, 4(1), 63-70.
- Zarco-Tejada et al. (2019). Chlorophyll content estimation in an open-canopy conifer forest with Sentinel-2A and hyperspectral imagery in the context of forest decline. *Remote sensing of environment*, 223, 320-335.
- Campbell et al. (2020). A multi-sensor, multi-scale approach to mapping tree mortality in woodland ecosystems. *Remote Sensing of Environment*, 245, 111853.
- Schiefer et al. (2023). UAV-based reference data for the prediction of fractional cover of standing deadwood from Sentinel time series. *ISPRS Open Journal of Photogrammetry and Remote Sensing*, 8, 100034.
- Furniss, T. J., Kane, V. R., Larson, A. J., & Lutz, J. A. (2020). Detecting tree mortality with Landsat-derived spectral indices: Improving ecological accuracy by examining uncertainty. *Remote Sensing of Environment*, 237, 111497.

REVIEWERS' COMMENTS

Reviewer #1 (Remarks to the Author):

Review of "Widespread scattered tree death contributes to substantial forest loss in California"

The authors have done an excellent and thorough job of responding to the comments by the reviewers. I appreciate the additional context and explanations in the adjusted text (main and supplementary) as well as in the response to reviews letter itself. I particularly applaud the authors for the additional temporal analysis (!!) which greatly strengthens an already strong paper. Very well done, and I look forward to seeing this work published.

Just a couple of minor points that stood out:

L122-123: I understand what you are saying here, but this reads a little awkwardly. Consider revising.

For the title, I might consider dropping "widespread" since I think it's implied in the "California" part. How does "Scattered tree death contributes to substantial forest loss in California" sound? Or "Scattered tree mortality contributes to substantial forest loss in California". I'll totally defer to the authors, but am flagging it since it jumped out at me.

Reviewer #2 (Remarks to the Author):

Overall, the revised manuscript greatly improved in quality and comprehensiveness. The authors have tackled all comments and suggestions and have been either integrated in the manuscript or sufficient explanations were given.

In my previous review, I highlighted that the research gap was unclear (from a methodological point of view). In the revised manuscript, the gap is now really crisp! Nicely done!

I am also happy that the authors followed the suggestion to apply a multitemporal tree mortality mapping. I think this greatly highlights the value of such technologies!

Fig. 2: the colors of the lines should be explained in the panel directly (e.g., with a legend) and not only in the caption. I think it is also a bit confusing what is shown here. The y-axis shows the amount of trees over the entire area, right? Then, I would express this in the y-axis title (e.g., dead tree count for the entire study region).

Fig. 4: I don't understand what the crosses mean. Is this for orientation? I think it will be rather confusing (my first idea was that it highlights the 30m grids, but it does not).

Line 326: You outline that small objects are less likely to be detected. Consider adding (if you have space left vs. word count) that this is a) a matter of object size vs. spatial resolution of the airborne data and b) what would be a solution to compensate for this issue, e.g., drone imagery or/and aerial imagery with higher resolution but also higher overlap to have better orthographic representation. In general, I think that you should provide some perspective on the limits and potentials of such technologies (e.g., integration with satellite data to look back in time or assess regions/years where we have no airborne data).

It would be of great value if you will provide your analytical tools in the form of a well-documented workflow to the community. Great work that will surely have a big impact!

Response to Reviewers

Note: To facilitate the reading of our answers to comments, changes of text are indicated in green. Unless it is stated, line numbers refer to the track changes version for easy referral.

+++++

Reviewer #1 (Remarks to the Author):

Review of "Widespread scattered tree death contributes to substantial forest loss in California"

The authors have done an excellent and thorough job of responding to the comments by the reviewers. I appreciate the additional context and explanations in the adjusted text (main and supplementary) as well as in the response to reviews letter itself. I particularly applaud the authors for the additional temporal analysis (!!) which greatly strengthens an already strong paper. Very well done, and I look forward to seeing this work published.

***Authors:** We thank the reviewer for acknowledging our work! We appreciate all the comments and suggestions which helped improve the manuscript.*

Just a couple of minor points that stood out:

L122-123: I understand what you are saying here, but this reads a little awkwardly. Consider revising.

***Authors:** We thank the reviewer for pointing this out. We revised the corresponding lines as follows (L129 - 131): "Tree mortality may be attributed to single or multiple damage agents^{12-14,22} depending on the interactions between damage agents such as a combination of biotic and abiotic agents or multiple species of pests or diseases."*

For the title, I might consider dropping "widespread" since I think it's implied in the "California" part. How does "Scattered tree death contributes to substantial forest loss in California" sound? Or "Scattered tree mortality contributes to substantial forest loss in California". I'll totally defer to the authors, but am flagging it since it jumped out at me.

***Authors:** We thank the reviewer for the suggestion. We have revised the title as "Scattered tree death contributes to substantial forest loss in California".*

Reviewer #2 (Remarks to the Author):

Overall, the revised manuscript greatly improved in quality and comprehensiveness. The authors have tackled all comments and suggestions and have been either integrated in the manuscript or sufficient explanations were given.

In my previous review, I highlighted that the research gap was unclear (from a methodological point of view). In the revised manuscript, the gap is now really crisp! Nicely done!

I am also happy that the authors followed the suggestion to apply a multitemporal tree mortality mapping. I think this greatly highlights the value of such technologies!

***Authors:** Many thanks to the reviewer for the comments and suggestions, which helped improve the manuscript!*

Fig. 2: the colors of the lines should be explained in the panel directly (e.g., with a legend) and not only in the caption. I think it is also a bit confusing what is shown here. The y-axis shows the amount of trees over the entire area, right? Then, I would express this in the y-axis title (e.g., dead tree count for the entire study region).

***Authors:** We thank the reviewer for the suggestion. We updated the figure and figure caption accordingly.*

Fig. 4: I don't understand what the crosses mean. Is this for orientation? I think it will be rather confusing (my first idea was that it highlights the 30m grids, but it does not).

***Authors:** We thank the reviewer for pointing this out. The crosses are for referencing the same location in scenes from different years. Due to geometrical and spectral variations in scenes from different years, it is difficult to pinpoint the same locations without these referencing crosses. Therefore, we decided to keep the referencing crosses and explained them in the caption as follows (L914 - 915): "The crosses in Panel d-g are for cross-referencing the same locations in scenes from different years."*

Line 326: You outline that small objects are less likely to be detected. Consider adding (if you have space left vs. word count) that this is a) a matter of object size vs. spatial resolution of the airborne data and b) what would be a solution to compensate for this issue, e.g., drone imagery or/and aerial imagery with higher resolution but also higher overlap to have better orthographic representation. In general, I think that you should provide some perspective on the limits and potentials of such technologies (e.g., integration with satellite data to look back in time or assess regions/years where we have no airborne data).

***Authors:** We thank the reviewer for the suggestion. We revised the corresponding line as follows (L281 - 283): "Higher resolution drone or/and aerial imagery³² and active remote sensing technologies such as Light Detection and Ranging (LiDAR) could compensate the mapping of small understory dead trees^{58,59}."*

It would be of great value if you will provide your analytical tools in the form of a well-documented workflow to the community. Great work that will surely have a big impact!

Authors: *We thank the reviewer for the suggestion. We have shared all scripts on figshare (i.e., data downloading, dead tree segmentation, post-processing, statistic analysis and visualization) with necessary annotations and as well as a README file describing the workflow.*